# Transcription factor IRF-5 regulates lipid metabolism and mitochondrial function in murine CD8⁺ T-cells during viral infection

Linh Thuy Mai[1], Sharada Swaminathan[1], Trieu Hai Nguyen[1], Etienne Collette[2], Tania Charpentier[1], Liseth Carmona-Pérez[1], Hamza Loucif[3,4,5,6], Alain Lamarre[1], Krista M Heinonen [ID][1], David Langlais [ID][2], Jörg H Fritz[2] & Simona Stäger [ID][1]✉

## Abstract

**Exhaustion of CD8⁺ T-cells leads to their reduced immune functionality and is controlled by numerous transcription factors. Here we show that the transcription factor IRF-5 helps to limit functional exhaustion of murine CD8⁺ T-cells during the chronic stage of LCMV (CL13) viral infection. Our results suggest that T-cell inhibitory receptors and transcription factor TOX, which are implicated in dampening T-cell activation and promoting exhaustion, are upregulated in infected IRF-5-deficient CD8⁺ T-cells. In addition, these cells display a reduced capacity to produce cytokines and lower survival rates than wild-type cells. Our findings indicate that these effects are mediated by defective lipid metabolism, increased lipid peroxidation, enhanced mitochondrial ROS production, and reduced levels of oxidative phosphorylation in the absence of IRF-5. These results identify IRF-5 as an important regulator of lipid metabolism and mitochondrial function that protects CD8⁺ T-cells from functional exhaustion during the chronic stage of viral infection.**

**Keywords** CD8 T Cells; IRF-5; Lipid Metabolism; Mitochondria; Cell Exhaustion
**Subject Categories** Immunology; Metabolism; Microbiology, Virology & Host Pathogen Interaction

## Introduction

Maintenance of CD8 T cells is essential for controlling infections caused by intracellular pathogens. However, CD8 T cell responses gradually fade during the chronic stages of infection and antigen-specific CD8 T cells become exhausted as a result of overwhelming antigenic stimulation (Moskophidis et al, 1993; Speiser et al, 2014; Zajac et al, 1998). Exhausted T cells (Tex) are characterized by an impaired proliferation, the progressive loss of effector functions, and the elevated expression of inhibitory receptors (Fuller and Zajac, 2003; Mackerness et al, 2010; McLane et al, 2019; Wherry et al, 2003). These changes are typically accompanied by a skewed cellular metabolism. At the early stages of exhaustion, Tex show a limited glucose uptake and a decline in glycolysis, which is ascribed to an elevated expression of PD-1 (Bengsch et al, 2016; Patsoukis et al, 2015). Defects in mitochondrial mass and fitness are also observed (Bengsch et al, 2016; Yu et al, 2020). These are attributed to the activation of Blimp-1, which suppresses the expression of PPAR-gamma co-activator 1α (PCG-1α), a key regulator of energy metabolism (Yu et al, 2020).

Several studies have reported a distinctive transcriptional profile for exhausted CD8 T cells. Indeed, numerous transcription factors have been shown to regulate the maintenance or differentiation of various exhausted CD8 T cell populations. For instance, during chronic infection with Lymphocytic choriomeningitis virus Clone 13 (LCMV Cl13), a high ratio of nuclear Eomes to T-bet positively correlates with the transcription of *Pdcd1* and the level of exhaustion (McLane et al, 2021). Moreover, T cell factor-1 (TCF-1) mediates the differentiation of exhausted CD8 T cell precursors (Utzschneider et al, 2016), negatively regulates the effector-like gene differentiation program, and is upstream of EOMES and c-Myb expression, which regulates cell survival (Chen et al, 2019b). Members of Nuclear factor of activated T cells (NFAT) family of transcription factors regulate the expression of inhibitory receptors thus promoting the Tex phenotype (Martinez et al, 2015; Oestreich et al, 2008). Recent studies have described thymocyte selection-associated high mobility group box (TOX) expression as a master mediator of T cells exhaustion for governing the expression of inhibitory receptors, exhaustion-associated markers, and EOMES and TCF-1 (Alfei et al, 2019; Khan et al, 2019; Scott et al, 2019; Seo et al, 2019; Yao et al, 2019). TOX expression is induced by persistent TCR activation and NFAT stimulation (Khan et al, 2019;

[1]Centre Armand-Frappier Santé Biotechnologie, Institut National de la Recherche Scientifique, Laval, QC, Canada. [2]Dahdaleh Institute of Genomic Medicine, Department of Microbiology and Immunology, Department of Human Genetics, McGill University, Montreal, QC, Canada. [3]Department of Microbiology and Immunology, McGill University, Montréal, QC, Canada. [4]FOCiS Centre of Excellence in Translational Immunology (CETI), Montréal, QC, Canada. [5]McGill University Research Centre on Complex Traits (MRCCT), Montréal, QC, Canada. [6]Department of Physiology, McGill University, Montreal, QC, Canada. ✉E-mail: simona.stager@inrs.ca

Scott et al, 2019). Other transcription factors, such as Nuclear Receptor Subfamily 4A (NR4A), Blimp-1, and Basic leucine zipper ATF-like Transcription Factor (BATF) also contribute to Tex development (Chen et al, 2019a; Jadhav et al, 2019; Quigley et al, 2010; Si et al, 2020). Recently, several members of the Interferon regulatory factor (IRF) family were also shown to regulate the CD8 T cell fate. While IRF-4 (Man et al, 2017) and IRF-2 (Lukhele et al, 2022) appear to be driving CD8 T cell exhaustion, IRF-9 seems to prevent it (Huber et al, 2017).

Interferon Regulatory Factor 5 (IRF-5), originally characterized as a transcription factor downstream of the Type I interferon signaling pathway in innate immune cells (Barnes et al, 2001), has been reported to play an essential role in tumor suppression (Bi et al, 2014), cell cycle regulation (Barnes et al, 2003), induction of apoptosis (Hu and Barnes, 2009), regulation of proinflammatory cytokines in myeloid lineages (Takaoka et al, 2005), M1 macrophage polarization (Krausgruber et al, 2011), and regulation of plasma cell development (De et al, 2017). Recently, IRF-5 was also shown to be downstream of TCR signaling in CD4 T cells (Yan et al, 2020), to regulates the induction of Th1- and Th17- associated cytokines and their migration (Yan et al, 2020), and to induce cell death in murine IFN-$\gamma^+$ CD4 T cells during chronic infection and in memory CD4 T cells of people living with HIV (Carmona-Perez et al, 2023; Fabie et al, 2018). Whether IRF-5 is expressed in CD8 T cells and the role it may have in these cells remains unknown.

Here we show that IRF-5 is upregulated and activated in CD8 T cells in mice infected with LCMV Cl13. In the absence of IRF-5, CD8 T cells enter the cell cycle in greater proportion; however, they fail to accumulate, display reduced survival capacity, and show increased signs of functional exhaustion during the chronic stage of infection, when compared with IRF-5 sufficient cells. Moreover, IRF-5 deficiency in CD8 T cells results in severely defective lipid metabolism, in an increased mitochondrial ROS production, and in the reduced capacity to produce ATP from oxidative phosphorylation. Finally, we demonstrate that IRF-5-deficiency in CD8 T cells also leads to increased lipid peroxidation and to cell death. Taken together, these results suggest a role for IRF-5 in limiting cell exhaustion and ultimately cell death during chronic LCMV infection by acting as a metabolic checkpoint in CD8 T cells, regulating mitochondrial remodelling and functions.

# Results

## IRF-5 is mostly expressed and activated in effector and memory-like CD8 T cells

IRF-5 has been shown to be expressed and activated in human and murine CD4 T cells (Carmona-Perez et al, 2023; Fabie et al, 2018); however, whether CD8 T cells also express IRF-5 remains unknown. Hence, we first monitored the expression of this transcription factor in murine CD8 T cells over the course of infection with LCMV Cl13, which establishes chronic infection in mice, and found that IRF-5 was indeed expressed by murine CD8 T cells (Fig. 1A). As already observed in CD4 T cells (Fabie et al, 2018), this transcription factor was mostly expressed in antigen-experienced cells, while only few cells from naive mice were IRF-5$^+$ (Fig. 1B,C). Interestingly, IRF-5 expression peaked around d30 p.i. in KLRG1$^+$ CD8 T cells, a time when cells start showing signs of

exhaustion. At this stage of infection, about 80% of the KLRG1$^+$ cells were IRF-5$^+$ (Fig. 1B). The kinetics of IRF-5 expression in CD127$^+$ CD8 T cells was slightly different. Indeed, higher frequencies of IRF-5$^+$ CD127$^+$ cells were observed at d8 and 34 p.i. (Fig. 1C). As IRF-5 dimerizes and translocate to the nucleus upon activation (Thompson et al, 2018), we next assessed the subcellular localization of this transcription factor. IRF-5 expression was found to colocalize with the nucleus in CD8 T cells during the whole course of infection, with 50–70% of the cells expressing IRF-5 in the nucleus between d8 and 60 p.i. (Fig. 1D). These results suggest that this transcription factor is expressed and is active in effector and memory-like CD8 T cells.

## IRF-5 is required to maintain CD8 T cells during chronic infection

To determine the function of IRF-5 in CD8 T cells, we generated IRF-5-deficient p14 transgenic mice (Irf5$^{flox/flox}$ x CMV-Cre$^+$ p14), which have CD8 T cells that are specific for the LCMV gp33 antigen. In these mice, IRF-5 ablation is ubiquitous. We first confirmed IRF-5 deletion in total splenocytes (Appendix Fig. S1A) and in splenic CD8 T cells (Appendix Fig. S1B) from Irf5$^{-/-}$ p14 mice by flow cytometry. Naive CD8 T cells purified from these mice and from the IRF-5 sufficient, Cre$^-$ p14 control mice (Irf5$^{flox/flox}$ x CMV-Cre$^-$ p14; herein called as WT p14) were then adoptively transferred into CD45.1 congenic mice a day prior to infection with LCMV Cl13 (Appendix Fig. S1C). Adoptive transfer experiments were carried out using separate groups of male and female mice. We found that male and female mice that received WT p14 cells had different survival rates (Appendix Fig. S1D), with only 60% of male mice surviving LCMV Cl13 infection compared to 80% of the females. However, the most remarkable difference was observed between male and female mice that received Irf5$^{-/-}$ p14 cells. While females adoptively transferred with Irf5$^{-/-}$ p14 cells survived infection to similar rate than those that received WT p14 cells, only 20% of male mice adoptively transferred with Irf5$^{-/-}$ p14 survived LCMV Cl13 infection at d15 p.i. (Appendix Fig. S1D). Because of these differences, we decided to characterize the function of IRF-5 in CD8 T cells from female and male mice separately. To this end, p14 cells were adoptively transferred into recipient mice and monitored at various time points over 60 days of infection. As expected, WT p14 cell expansion peaked at d8 p.i. to slightly contract during the chronic stage of infection (Fig. 2A). This kinetics was reflected in the percentages of p14 cells (Fig. 2B) as well as in the p14 cell numbers (Fig. 2C) observed over the course of infection. Surprisingly, IRF-5-deficient p14 cells underwent clonal expansion similarly to WT p14 cells but failed to survive during the chronic stage of infection (Fig. 2A). Indeed, a dramatic decrease in cell frequencies (Fig. 2B) and numbers (Fig. 2C) was observed at d30 and 60 p.i. in mice that received IRF-5-deficient p14 cells compared to the control group. Interestingly, IRF-5 deficiency mostly impacted KLRG1$^+$ cells, which were present at significantly lower frequencies in the spleen of mice that received Irf5$^{-/-}$ p14 compared to WT p14 cells at d30 and 60 p.i. (Fig. 2D,E). In contrast, noticeably higher frequencies of KLRG1$^+$ cells were observed at d8 p.i. in the spleen following adoptive transfer of Irf5$^{-/-}$ p14 compared to WT p14 cells (Fig. 2D,E). CD127$^+$ memory-like cells were less affected by the lack of IRF-5 (Fig. 2F). We found similar results in female mice, where IRF-5-deficiency

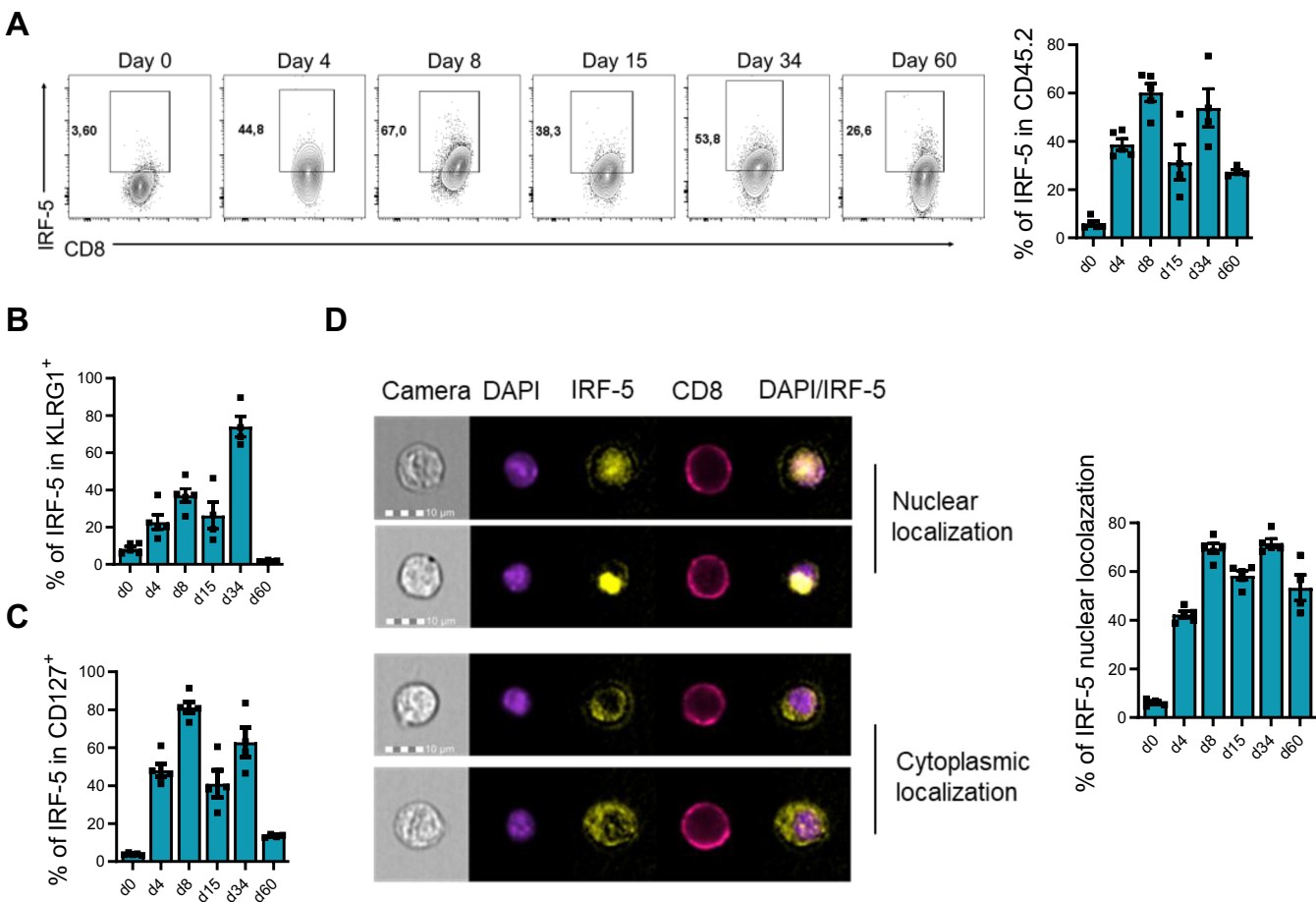

**Figure 1. IRF-5 is expressed in both KLRG1$^+$ and CD127$^+$ CD8 T cells in LCMV Cl13-infected mice.**

Sorted naive p14 CD8 T cells were transferred into CD45.1 recipient mice one day prior to intravenous infection with $2 \times 10^6$ PFU LCMV Cl13. Mice were euthanized at various time points after infection. Cells were gated on CD45.2$^+$ CD8$^+$; graphs show (A) representative flow cytometry plots (left) and the frequency (right) of p14 CD8 T cells expressing IRF-5 present in the spleen of infected mice; (B) the percentage of KLRG1$^+$ CD127$^-$ p14 CD8 T cells expressing IRF-5; (C) the percentage of CD44$^+$ CD127$^+$ p14 CD8 T cells expressing IRF-5; and (D) the frequency of IRF-5 expression that colocalizes with DAPI staining over the course of LCMV Cl13 infection. Data represent the mean ± SEM of one of two or three independent experiments, $n = 4-5$.

resulted in higher percentages of KLRG1$^+$ cells at d8 p.i. compared with WT p14 cells but KLRG1$^+$ responses were not maintained to the same level in the absence of IRF-5 (Appendix Fig. S1E); moreover, the frequencies of CD127+ cells were not affected by the absence of IRF-5 (Appendix Fig. S1F).

Taken together, these results suggest that IRF-5 deficiency in p14 cells results in $Irf5^{-/-}$ p14 cells undergoing similar clonal expansion than WT p14 cells but then failing to survive during the chronic phase of infection.

To confirm our results, we infected $Irf5^{flox/flox}$ x $Lck-Cre^+$ and $Cre^-$ mice (herein referred to as $Lck-Cre^+$ and WT) with LCMV Cl13 and monitored CD8 T cell responses over the course of infection. A more pronounced decline in percentages (Appendix Fig. S2A,B) and numbers (Appendix Fig. S2C) of total CD8 T cells was observed from d28 p.i. in mice with a T cell-specific IRF-5 ablation compared with the control group, indicating that IRF-5 deficiency affected endogenous, non-transgenic cell survival as well. Similar results were obtained when we followed gp$_{33-41}$ tetramer positive cells (Appendix Fig. S2D–F).

### $Irf5$ ablation results in functional exhaustion of CD8 T cells

We next investigated whether the absence of IRF-5 in p14 cells affected their effector function. Therefore, the production of various cytokines was assessed at different time points of infection upon restimulation. We found that $Irf5^{-/-}$ p14 cells have a superior capacity to generate IFN-γ-producing cells at d8 p.i. in terms of frequencies (Fig. 3A,B, upper panel) and cytokine amount (Fig. 3B, lower panel), compared with WT p14 cells. Similar results were observed for TNF- (Fig. 3A,C) and INF-γ-TNF-double producing cells (Fig. 3A,D), reflecting our previous observation that mice that received $Irf5^{-/-}$ p14 displayed higher frequencies of KLRG1$^+$ effectors at d8 p.i. (Fig. 2E). Nevertheless, the absence of IRF-5 significantly impacted the maintenance of IFN-γ-producing cells over the chronic stage of infection, when lower frequencies of IFN-γ$^+$ cells (Fig. 3A,B, upper panel) and lower amounts of IFN-γ per cell (Fig. 3B, lower panel) were observed, compared with the control group. The percentage of TNF$^+$ (Fig. 3A,C) and IFN-γ $^+$TNF$^+$

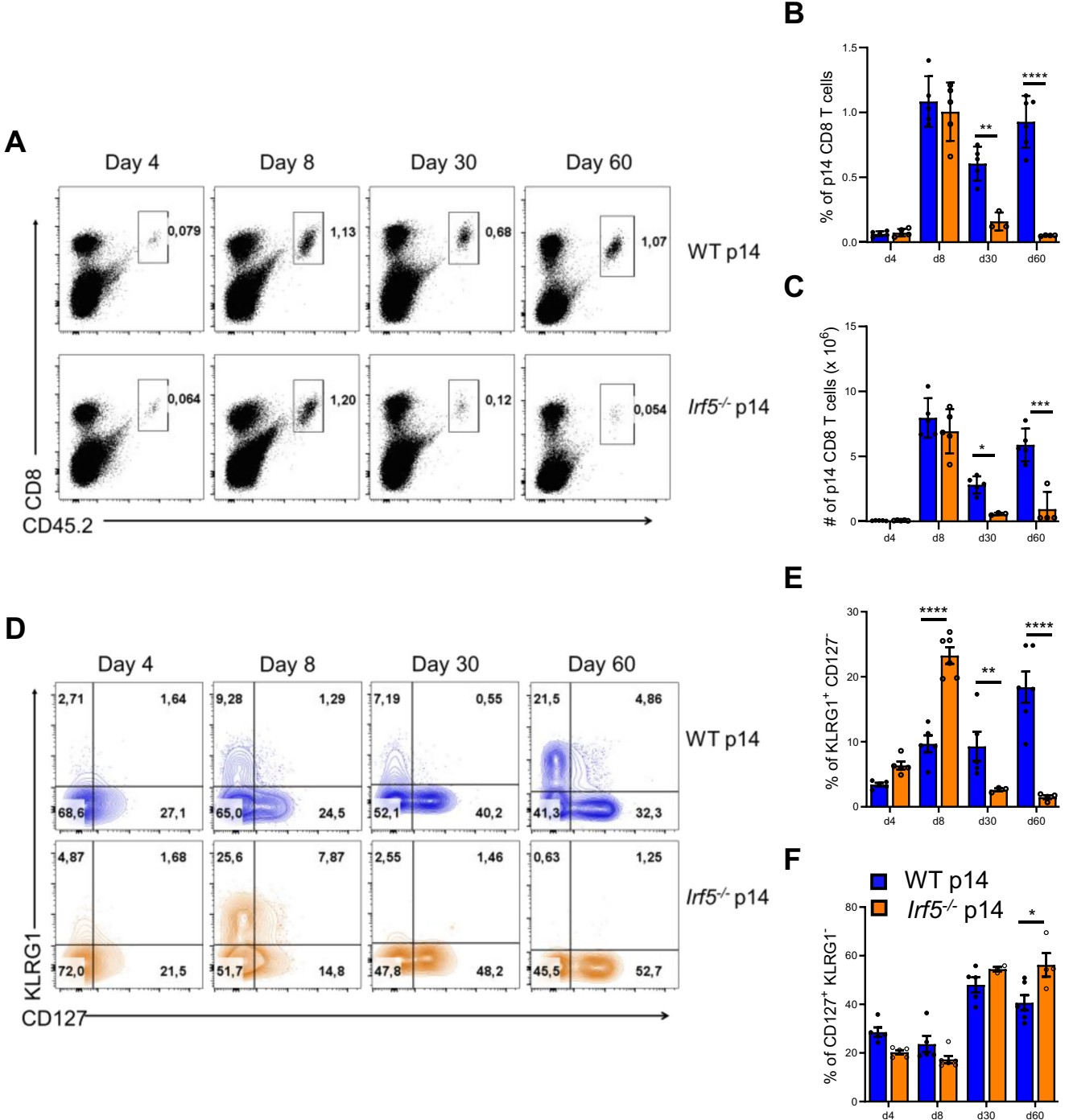

**Figure 2. IRF-5 is required to maintain CD8 T cell responses throughout infection.**

CD8 T cells from male WT or *Irf5*−/− p14 mice were transferred into CD45.1 mice one day prior to intravenous infection with 2 × 10⁶ PFU LCMV Cl13. Mice were euthanized at various time points after infection. Graphs show (**A**) representative FACS plots for splenocytes stained with anti-CD8 and anti-CD45.2; the percentage (**B**) and the absolute numbers (**C**) of adoptively transferred WT and *Irf5*−/− p14 CD8 T cells present in the spleen of infected recipient mice; (**D**) representative flow cytometry plots and corresponding percentages of KLRG1⁺ CD127⁻ (**E**) and CD127⁺ CD44⁺ (**F**) p14 CD8 T cells in the spleen of recipient mice over the course of LCMV infection. *P* values from left to right (**B**): *P* = 0.0064, *P* = 6.56E-09. *P* values from left to right (**C**): *P* = 0.033, *P* = 4.95E-06. *P* values from left to right (**E**): *P* = 8.15E-06, *P* = 0.049, *P* = 4.07E-07, (**F**) *P* = 0.011. Statistical significance was determined by two-way ANOVA. Data represent the mean ± SEM, *n* = 4–5, one of three independent experiments is shown. *P < 0.05, **P < 0.01 ***P < 0.001 ****P < 0.0001.

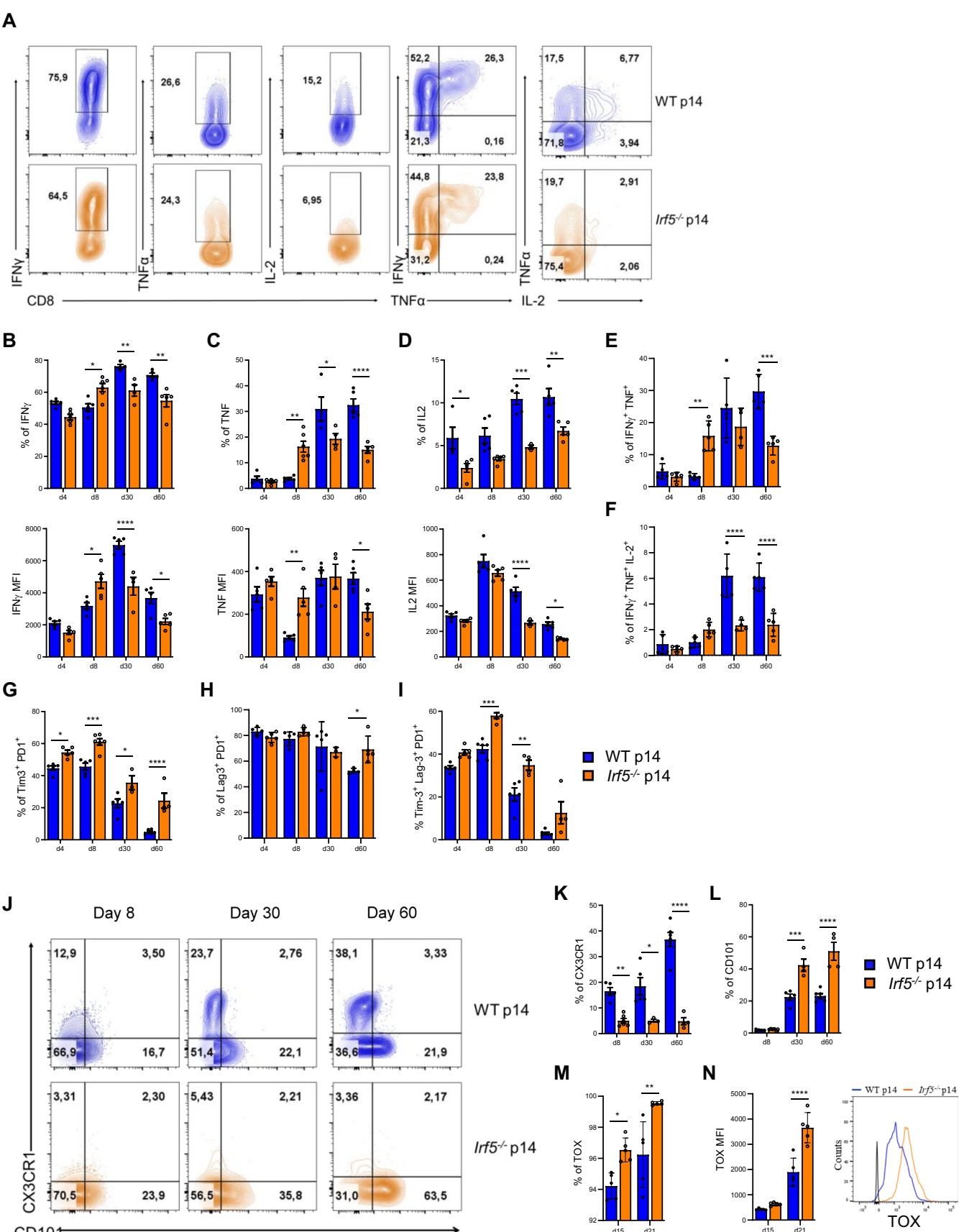

◀

**Figure 3. IRF5 ablation is associated with functional exhaustion during chronic infection.**

CD8 T cells from male WT or *Irf5*$^{-/-}$ p14 mice were adoptively transferred into CD45.1 recipient mice one day prior to intravenous infection with $2 \times 10^6$ PFU LCMV Cl13. Mice were euthanized at various time points after infection. (**A–F**) Cytokine production by adoptively transferred p14 cells was assessed after restimulation with the gp33 peptide. (**A**) Representative FACS plot for the various cytokine staining at d30 p.i. (**B–D**) Graphs show the frequency (upper graphs) and mean fluorescence intensity (MFI) (lower graphs) of IFN-γ$^+$ (**B**), TNF$^+$ (**C**), and IL-2$^+$ (**D**) WT p14 and *Irf5*$^{-/-}$ p14 CD8 T cells at various days post infection. (**E, F**) Graphs represent the percentage of IFN-γ$^+$ TNF$^+$ (**E**) and IFN-γ$^+$ TNF$^+$ IL-2$^+$ (**F**) WT p14 and *Irf5*$^{-/-}$ p14 CD8 T cells at various days post infection. *P* values from left to right (**B**, upper graph): $P = 0.016$, $P = 0.0097$, $P = 0.0011$. *P* value from left to right (**B**, lower graph): $P = 0.023$, $P = 5.24\text{E-}05$, $P = 0.037$. *P* values from left to right (**C**, upper graph): $P = 0.0019$, $P = 0.025$, $P = 2.41\text{E-}05$. *P* values from left to right (**C**, lower graph): $P = 0.0017$, $P = 0.011$. *P* values from left to right (**D**, upper graph): $P = 0.048$, $P = 0.0009$, $P = 0.0098$. *P* values from left to right (**D**, lower graph): $P = 4.84\text{E-}05$, $P = 0.012$. *P* values from left to right (**E**): $P = 0.005$, $P = 0.00011$. *P* values from left to right (**F**): $P = 7\text{E-}06$, $P = 5.05\text{E-}06$. (**G–I**) Expression of inhibitory receptors in WT and *Irf5*$^{-/-}$ p14 CD8 T cells. Graphs represent the frequency of Tim3$^+$ PD1$^+$ (**G**), Lag3$^+$ PD1$^+$ (**H**), and Tim3$^+$ Lag3$^+$ PD1$^+$ (**I**) adoptively transferred WT p14 and *Irf5*$^{-/-}$ p14 cells over the course of infection. *P* values from left to right (**G**): $P = 0.036$, $P = 0.00069$, $P = 0.036$, $P = 5.18\text{E-}05$. (**H**): $P = 0.032$. *P* values from left to right (**I**): $P = 0.0002$, $P = 0.0045$. (**J–L**) Representative flow cytometry plots and graphs showing the percentage of CX3CR1$^+$ effector-like (**J, K**) and CD101$^+$ terminally exhausted (**J, L**) WT and *Irf5*$^{-/-}$ p14 CD8 T cells at various time points post infection. (**M, N**) Graphs represent the percentage (**M**) and intensity (**N**) of TOX expression at various days post infection. *P* values from left to right (**K**): $P = 0.0084$, $P = 0.011$, $P = 8.38\text{E-}09$. *P* values from left to right (**L**): $P = 0.00013$, $P = 3.46\text{E-}07$. *P* values from left to right (**M**): $P = 0.035$, $P = 0.0026$. $P = 2.37\text{E-}5$ (**N**). Statistical significance was determined by two-way ANOVA. Data represent the mean ± SEM, $n = 4–5$, one of three independent experiments is shown, *$P < 0.05$, **$P < 0.01$, ***$P < 0.001$, ****$P < 0.0001$.

(Fig. 3A,E) *Irf5*$^{-/-}$ p14 cells was also significantly lower than the control group at d30 and d60 p.i. Interestingly, mice that received *Irf5*$^{-/-}$ p14 cells showed reduced frequencies of IL-2-producing cells over the whole course of infection compared with those that were adoptively transferred with WT p14 (Fig. 3A,D, upper panel). Moreover, *Irf5*$^{-/-}$ p14 cells produced less IL-2 than WT p14 cells (lower panel) and dramatically lower percentages of IFN-γ$^+$ TNF$^+$ IL-2$^+$ were observed for *Irf5*$^{-/-}$ p14 compared with WT p14 cells (Fig. 3A,F). Similar results were obtained during the chronic phase of infection using female mice (Appendix Fig. S3A–C).

Because these findings suggest that *Irf5*$^{-/-}$ p14 cells are more dysfunctional at d30 and 60 of infection than their IRF-5-sufficient counterpart, we next assessed whether these cells would show increased signs of exhaustion. First, we assessed the expression of three inhibitory receptors that are typically expressed by exhausted CD8 T cells, namely PD-1, TIM-3, and LAG-3 (Wherry et al, 2007). We observed significantly higher frequencies of cells co-expressing PD-1 and TIM-3 in IRF-5-deficient p14 cells over the entire course of infection, compared with the control group (Fig. 3G). Percentages of PD-1$^+$ LAG-3$^+$ cells were very similar in both study groups with exception of d60 p.i., when these cells' frequency was higher in the *Irf5*$^{-/-}$ p14 group compared with the control group (Fig. 3H). However, triple positive cells (PD-1$^+$ Tim-3$^+$ Lag-3$^+$) were significantly more present in the absence of IRF-5 (Fig. 3I), suggesting that IRF-5 deficiency could worsen CD8 T cell exhaustion. To confirm this possibility, we assessed cell surface modulation of the chemokine receptor CX3CR1 and the transmembrane glycoprotein CD101. These two markers allow to differentiate exhausted (CD101$^+$) from effector-like (CX3CR1$^+$) CD8 T cells during chronic infections (Hudson et al, 2019). As expected, we found dramatically reduced frequencies of CX3CR1$^+$ effector-like cells (Fig. 3J,K) and a significantly higher percentage of CD101$^+$ exhausted cells (Fig. 3J,L) in the absence of IRF-5 when compared with the IRF-5 sufficient group. Similar results were obtained using female mice (Appendix Fig. S3D,E). Moreover, IRF-5-deficient p14 cells expressed TOX, a transcription factor known to drive CD8 T cell exhaustion (Khan et al, 2019; Scott et al, 2019), at higher frequencies (Fig. 3M) and levels (Fig. 3N) compared with WT p14. We confirmed these results using a different set of markers to distinguish progenitor-like cells (Tex$^{prog1}$ and Tex$^{prog2}$) from exhausted intermediate (Tex$^{int}$) and terminally exhausted cells

(Tex$^{term}$), as described by Beltra et al (Beltra et al, 2020) (Appendix Fig. S3F–J) and observed a dramatic reduction in the frequencies of Ly108$^-$CD69$^-$ Tex$^{int}$ cells (Appendix Fig. S3F,G) and a significant increase in the percentage of Ly108$^-$CD69$^+$ Tex$^{term}$ cells (Appendix Fig. S3F,H) in the *Irf5*$^{-/-}$ p14 group compared with WT p14 throughout the course of infection. Frequencies of Ly108$^+$CD69$^+$ Tex$^{prog1}$ (Appendix Fig. S3F,I) and Ly108$^+$CD69$^-$ Tex$^{prog2}$ (Appendix Fig. S3F,J) were similar between the two groups with exception of d60 p.i., when mice adoptively transferred with *Irf5*$^{-/-}$ p14 cells showed significantly higher percentages of Tex$^{prog1}$ cells.

IRF-5 was shown to compete for DNA binding with IRF-4 (Negishi et al, 2005), which in turn is known to drive CD8 T cell exhaustion (Man et al, 2017). Thus, we next assessed whether the absence of IRF-5 expression affected at all IRF-4 expression. Interestingly, we did not observe any differences in the frequency of CD8 T cells expressing IRF-4 between the two groups at any time points observed (Appendix Fig. S4A). Likewise, the proportion of CD8 T cell expressing Eomes (Appendix Fig. S4B) and T-bet (Appendix Fig. S4C) was not altered in the absence of IRF-5, except for d30 p.i., when *Irf5*$^{-/-}$ p14 cells expressed significantly lower levels of T-bet compared with their WT counterparts (Appendix Fig. S4C).

We also analyzed effector function and inhibitory receptor expression in T cell-specific *Irf5*$^{-/-}$ mice infected with LCMV Cl13 and found that in the absence of IRF-5, stronger signs of functional exhaustion (Appendix Fig. S5A–C) and higher frequencies of inhibitory receptors' expressions (Appendix Fig. S5D–F) were observed in CD8 T cells, compared with the control group. Interestingly, mice with a T cell specific *Irf5* ablation failed to clear infection in the serum (Appendix Fig. S5G), highlighting the severe CD8 T cell dysfunctionality.

## IRF-5 deficiency alters the expression of genes involved in the cell cycle

To further characterize the role of IRF-5 in CD8 T cells, we compared the transcriptomic profile of IRF-5-sufficient and -deficient p14 cells by bulk RNA sequencing. To this end, we performed adoptive transfer experiments as described in Appendix Fig. S1C, using male and female mice. Adoptively transferred cells were purified from the spleen of mice infected with LCMV Cl13 at d21 p.i. and bulk RNA sequencing was performed on two groups of

male and female mice, representing a pool of 7-17 mice each. Surprisingly, we only found few genes that were significantly differentially expressed. The top up- and downregulated genes are shown in Appendix Table S1. We first performed gene set enrichment analysis (GSEA) of transcription signatures from the Molecular Signature Database (MSigDB). We found a strong enrichment score for genes regulating the cell division (Fig. 4A) and cell cycle process (Fig. 4B) in IRF-5-deficient p14 cells compared with WT p14 cells. These results were not surprising, given the known role of IRF-5 in cell cycle arrest (Barnes et al, 2003). Hence, we decided to validate these findings in vivo and analyze the proportion of adoptively transferred IRF-5-sufficient and -deficient p14 cells that were in the G0, G1, and S-G2-M phase at d4, 8, 30, and 60 after infection with LCMV Cl13. We observed a significantly higher proportion of $Irf5^{-/-}$ p14 cells that were in the G1 and S-G2-M phase at d30 and 60 p.i., when compared with WT p14 cells (Fig. 4C). This suggests that $Irf5^{-/-}$ p14 cells were indeed entering the cell cycle in greater proportion. To monitor whether this would also occur in vitro, we stimulated purified naive $Irf5^{-/-}$ and WT p14 cells in vitro with the gp33 peptide and monitored the proportion of cells in the G0, G1 and S-G2-M phase 6, and 24 h after stimulation. No differences were observed between both groups in the absence of stimulation (Fig. 4D). As expected, $Irf5^{-/-}$ p14 cells were entering the cell cycle in greater proportion than their WT counterpart already at 6 h after stimulation with the gp33 peptide in vitro as well; this difference was sustained over the first 24 h of culture (Fig. 4D). This suggests that the absence of IRF-5 leads to enhanced cell division. Nevertheless, even though they enter the cell cycle more rapidly, the proportion and numbers of $Irf5^{-/-}$ p14 cells found in the spleen of LCMV Cl13-infected mice at d30 and 60 p.i. is significantly lower than WT p14 cells (Fig. 2B,C).

## IRF-5 deficiency in CD8 T cells affects the lipid metabolism

To better understand why $Irf5^{-/-}$ p14 cells fail to accumulate, we went back to our RNA sequencing data and assessed whether cell exhaustion pathways (as described by Wherry et al (Wherry et al, 2007)) were enhanced in $Irf5^{-/-}$ compared with WT p14 cells. We found a strong enrichment of signature genes for T cell exhaustion (Appendix Fig. S6), confirming our in vivo results (Fig. 3; Appendix Fig. S3). Thus, we reasoned that $Irf5^{-/-}$ p14 cells fail to accumulate because they were dying. Indeed, significantly higher frequencies of Annexin V-expressing cells (Fig. 5A) and intensities of Annexin V staining (Fig. 5B) were observed at d15 and 21 after LCMV Cl13 infection in adoptively transferred $Irf5^{-/-}$ p14 compared with WT p14 cells. Surprisingly, we did not find major differences in the frequency of cleaved Caspase 3$^+$ cells (Fig. 5C) or in the intensity of expression of cleaved Caspase 3 (Fig. 5D) between both groups, with exception of d21, when the percentage of cleaved Caspase 3$^+$ $Irf5^{-/-}$ p14 cells was slightly higher than their WT counterpart (Fig. 5C). These results suggest that, although cell death occurs more prominently in $Irf5^{-/-}$ p14 cells, this is not only caused by an increase in apoptotic cell death but could result from other cell death pathways as well.

Hence, we next sought to identify possible pathways that could compromise cell survival. Because IRF-5 was shown to be involved in mitochondrial function and cellular metabolism (Albers et al, 2021; Orliaguet et al, 2022), we first assessed the enrichment

signature of gene sets related to cellular metabolism and mitochondria. We obtained a negative enrichment score for mitochondrial gene expression (Fig. 5E) and genes involved in the regulation of the cellular respiration (Fig. 5F) in $Irf5^{-/-}$ compared with WT p14 cells, suggesting a defect in mitochondrial function and energy production in these cells. To validate these results, we assessed the mRNA expression levels of Coq4, Ndufaf3, and Ndufb4 in WT and $Irf5^{-/-}$ p14 cells purified ex vivo from female LCMV Cl13-infected mice at d21 p.i. Coq4 encodes for the coenzyme Q4, a protein involved in the biosynthesis of an essential component of the electron transport chain in mitochondria (Brea-Calvo et al, 2015); Ndufaf3 encodes for the NADH hydrogenase 1 alpha subcomplex assembly factor, a protein involved in the assembly and function of the mitochondria respiratory chain complex I (Saada et al, 2009); Ndufb4 encodes for the NADH hydrogenase 1 beta subcomplex 4, a subunit of the mitochondrial respiratory chain complex I (Wu et al, 2016). We found that indeed IRF-5-deficient p14 cells expressed significantly lower levels of those three genes when compared with IRF-5-sufficient p14 cells (Fig. 5G). A recent study by Orliaguet et al, has shown that IRF-5 regulates the mitochondrial architecture remodelling by transcriptionally repressing a key mitochondrial component for oxidative respiration, namely GHITM (Orliaguet et al, 2022). This work was done in macrophages and in the context of a high-fat diet. When we analyzed Ghitm mRNA levels in ex vivo purified cells, we observed a significant upregulation of this gene's expression in IRF-5-deficient compared with IRF-5-sufficient cells (Fig. 5H).

Mitochondria are the cellular compartment where fatty acid metabolism occurs; thus, we also assessed whether this latter was affected in the absence of IRF-5. We found a downregulation in the transcriptional signature of the fatty acid catabolic process (Fig. 5I) and the fatty acid beta oxidation (Fig. 5J) in $Irf5^{-/-}$ cells. To confirm these results, we assessed the expression of Gpat2, Awat2, and Gpdl1 mRNA levels in WT and $Irf5^{-/-}$ p14 cells purified ex vivo from the spleen of female mice at d21 p.i. Gpat2 encodes for glycerol-3-phosphate acyltransferase 2, an enzyme involved in triacylglycerol (TAG) synthesis (Cattaneo et al, 2012); Awat2 encodes for acyl-CoA wax alcohol acyltransferase 2, an enzyme that catalyzes the synthesis of wax esters (Arne et al, 2017); and Gpdl1 encodes for glycosylphosphatidylinositol-specific phospholipase D, an enzyme involved in triglyceride metabolism. We found that $Irf5^{-/-}$ p14 cells had a significantly lower expression of Gpat2, Awat2, and Gpdl1 mRNA levels when compared with the control group (Fig. 5K). Taken together, these results suggest that IRF-5-deficient p14 cells may undergo cell death during LCMV Cl13 infection because of a defect in their mitochondrial function and lipid metabolism.

## IRF-5 deficiency profoundly affects cell respiration and ATP production

We next analyzed the bioenergetic flux in live cells to investigate the oxidative metabolism of female WT and $Irf5^{-/-}$ p14 cells stimulated or not with the gp33 peptide or anti-CD3/CD28. Thus, we quantified the oxygen consumption rate (OCR) of both groups of cells at 6 (Fig. 6A) and 24 h (Fig. 6B) after stimulation with the gp33 peptide or anti-CD3/CD28. We found profound defects in the respiratory capacity of IRF-5-deficient p14 cells, which had a severely compromised basal (Fig. 6B,C) and maximal (Fig. 6B,D)

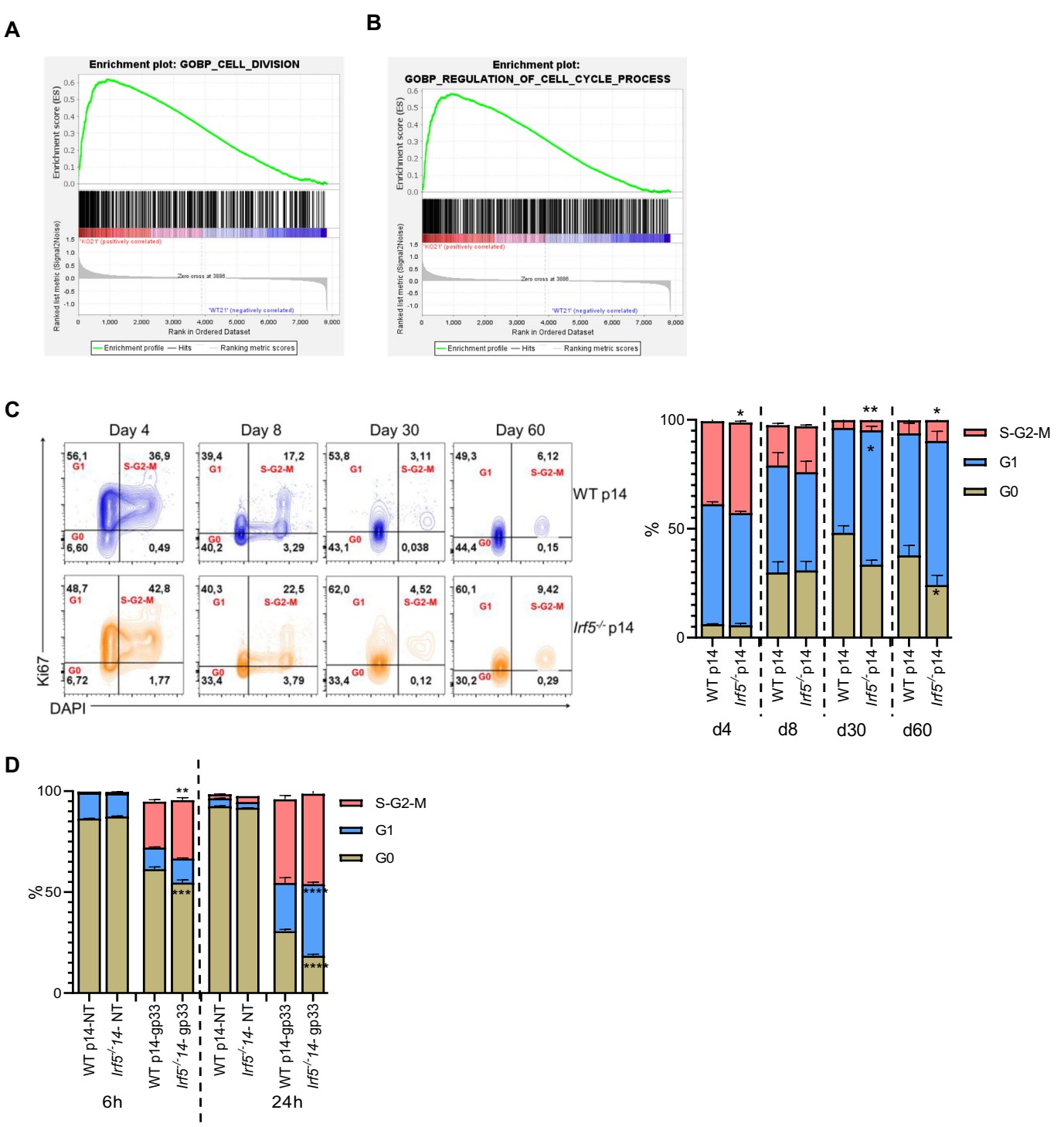

**Figure 4. IRF-5-deficiency alters the expression of genes involved the cell cycle.**

(A, B) Bulk RNA sequencing was performed on adoptively transferred WT p14 and *Irf5⁻/⁻* p14 CD8 T cells isolated from male and female recipient mice at day 21 after LCMV Cl13 infection.- (A, B) GSEA results showing enrichment of genes involved in cell division (A) and the cell cycle regulation (B) in *Irf5⁻/⁻* p14 cells compared with WT cells. (C) WT and *Irf5⁻/⁻* p14 CD8 T cells were adoptively transferred into CD45.1 recipient mice a day prior to infection with LCMV Cl13. Mice were then euthanized at various time point after infection. Graphs show representative flow cytometry plots for Ki67 and DAPI staining (upper graphs) and the proportion of cells in the various phases of the cell cycle at any given time point (lower graph). (D) Graph displays the cell cycle analysis based on Ki67 and DAPI staining of WT and *Irf5⁻/⁻* p14 CD8 T cells stimulated in vitro with or without the gp33 peptide. Statistical significance was determined by two-way ANOVA. *P* value day 4 (C): *P* = 0.017. *P* values day 30 from top to bottom (C): *P* = 0.0041, *P* = 0.019. *P* values day 60 from top to bottom (C): *P* = 0.003, *P* = 0.024. *P* values 6 h from top to bottom (D): *P* = 0.0035, *P* = 0.0002. *P* values 24 h from top to bottom (D): *P* = 3.03E-05, *P* = 2.25E-06. Data represent the mean ± SEM, *n* = 4–5, one of three independent experiments is shown, **P* < 0.05, ***P* < 0.01.

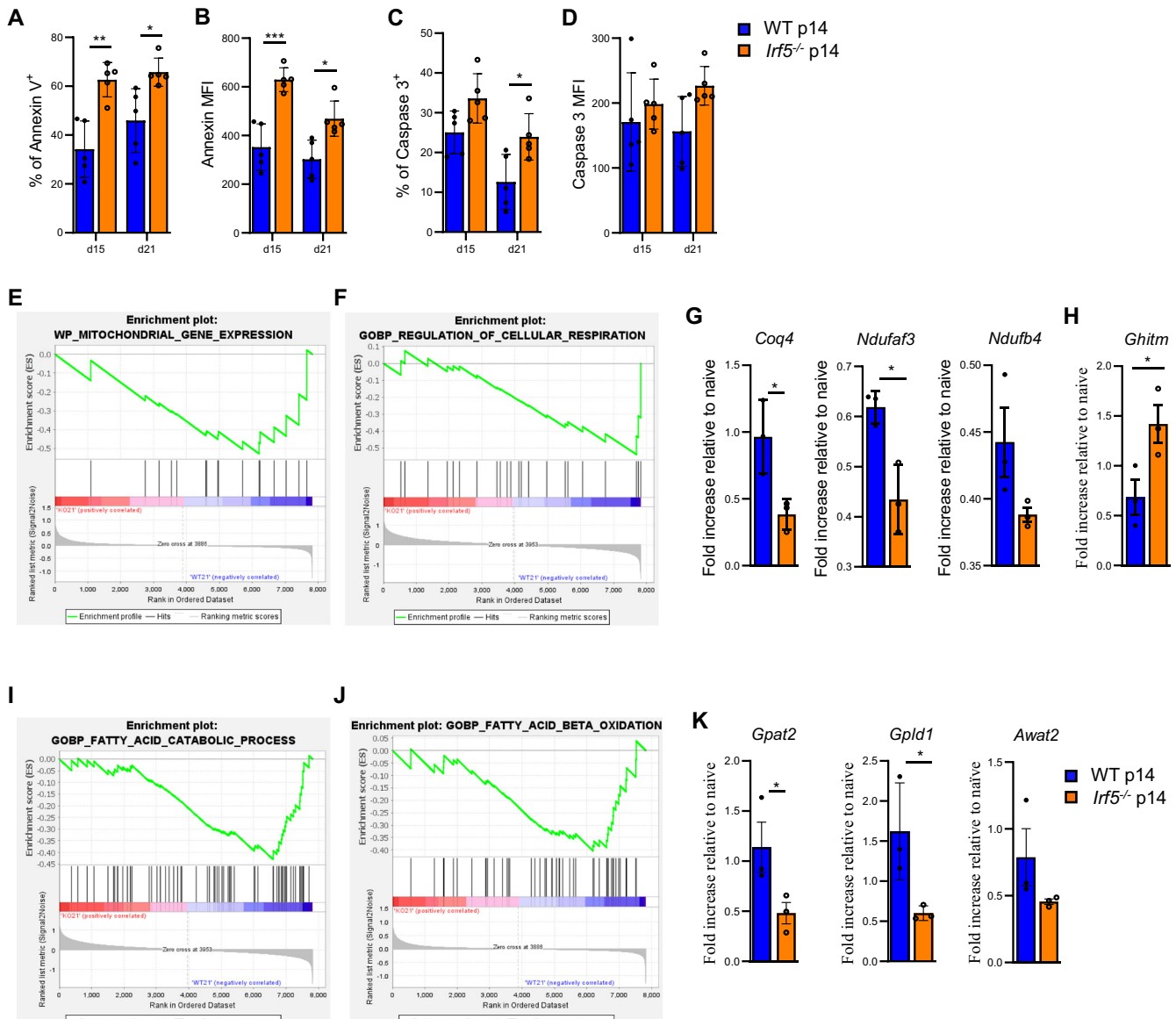

**Figure 5. IRF-5-deficiency affects the expression of genes involved in the lipid metabolism.**

(A–D) WT and *Irf5⁻/⁻* p14 CD8 T cells were adoptively transferred into female CD45.1 recipient mice a day prior to infection with LCMV Cl13. Mice were euthanized at d15 and 21 p.i. Graphs show the percentage of Annexin V⁺ (A), the mean fluorescence intensity of Annexin V staining (B), the percentage of cleaved Caspase 3 (C) and the mean fluorescence intensity of cleaved Caspase 3 staining (D). *P* values from left to right (A): *P* = 0.0016, *P* = 0.026. *P* values from left to right (B): *P* = 0.00014, *P* = 0.015. *P* = 0.043 (C). Statistical significance was determined by two-way ANOVA. Data represents the mean ± SEM, *P < 0.05, **P < 0.01, ***P < 0.001. Transcriptional profiling was performed on adoptively transferred WT p14 and *Irf5⁻/⁻* p14 CD8 T cells isolated from male and female recipient mice at day 21 post LCMV Cl13 infection. (E, F) Gene enrichment analyses show a negative enrichment score for genes involved in mitochondrial gene expression (E) and the regulation of cellular respiration (F). WT and *Irf5⁻/⁻* p14 CD8 T cells were adoptively transferred into female CD45.1 recipient mice a day prior to infection with LCMV Cl13. (G, H) Adoptively transferred cells were then sorted at d21 p.i. and the mRNA expressions of (G) *Coq4* (left graph), *Ndufaf3* (center graph), and *Ndufb4* (right graph), and (H) *Ghitm* were measured by digital droplet PCR. (I, J) Gene enrichment analyses showing enrichment scores for fatty acid catabolism (I), and fatty acid beta oxidation (J). (K) Adoptive transfer experiments were performed as described above and mRNA expressions of *Gpat2* (left graph), *Gpld1* (center graph), and *Awat2* (right graph) were analyzed by digital droplet PCR. *P* values from left to right (G): *P* = 0.028, *P* = 0.014. *P* = 0.047 (H). *P* values from left to right (K): *P* = 0.042, *P* = 0.044. Statistical significance was determined by *t* test. Data represent the mean ± SEM, *P < 0.05.

respiratory capacity. The ATP production derived from OCR was also dramatically reduced compared with WT p14 cells (Fig. 6E). We also assessed the extracellular acidification rate (ECAR) of both groups of cells at 6 and 24 h after stimulation with the gp33 peptide or anti-CD3/CD28 (Appendix Fig. S7A,B). With exception of the ECAR from IRF-5 deficient cells stimulated with gp33 that was

slightly higher than the other groups at 6 h after stimulation, *Irf5⁻/⁻* p14 cells exhibited lower ECAR that their WT counterpart. When we analyzed the bioenergetic profile at 6 and 24 h after stimulation, we found that WT p14 cells stimulated with the gp33 peptide or anti-CD3/CD28 could gain ATP from both mitochondrial respiration and glycolysis, whereas IRF-5-deficient p14 cells were nearly

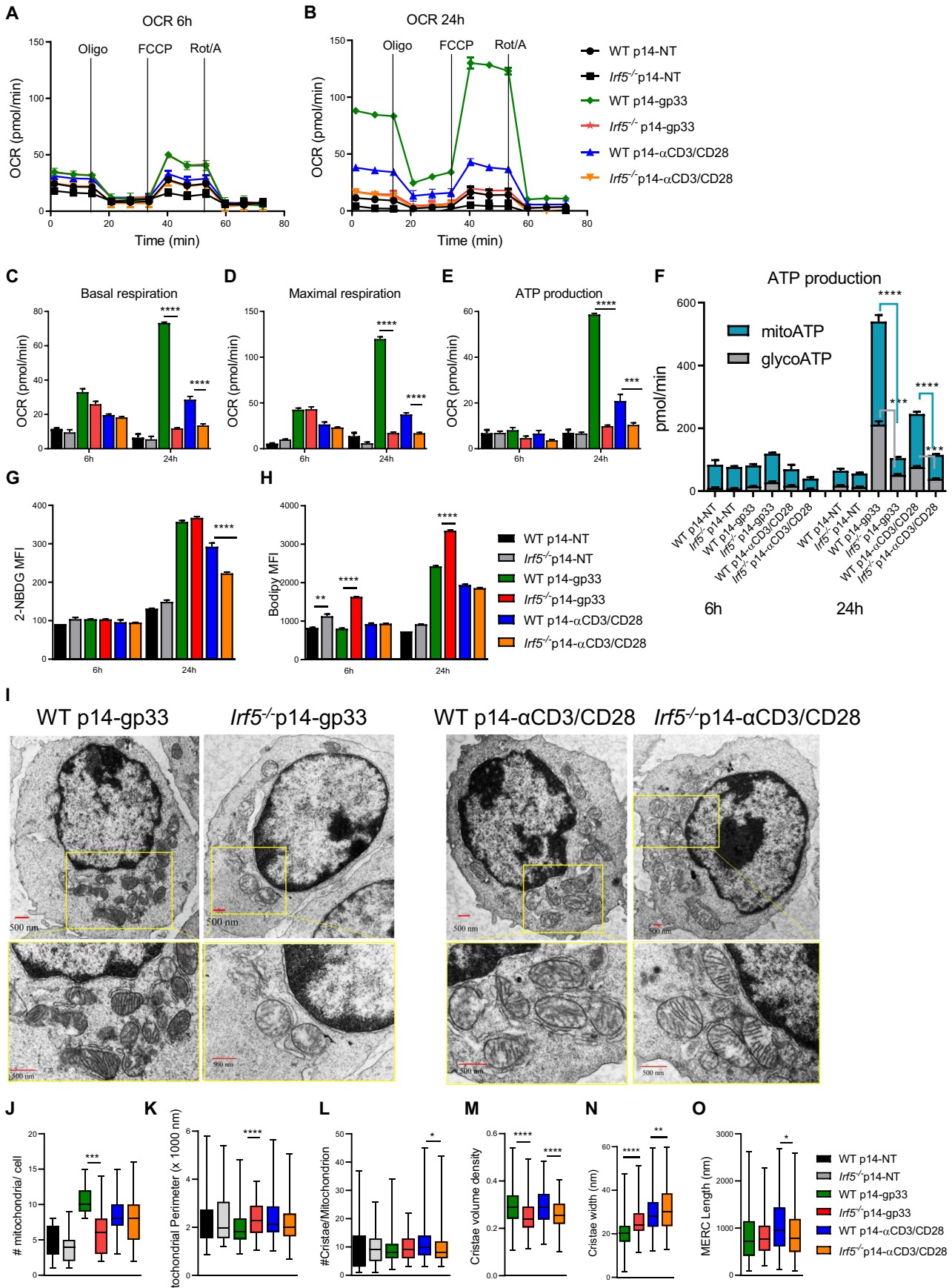

**Figure 6. IRF-5-deficient p14 cells show profound defects in the cellular respiration and ATP production.**

WT p14 and *Irf5*$^{-/-}$ p14 CD8 purified from female mice were cultured in vitro in the presence or absence of the gp33 peptide or anti-CD3/CD8, and the mitochondria respiratory capacity was measured using the Seahorse XFe-96 analyzer. Graphs illustrate the representative oxygen consumption rates (OCR) over time after 6 h (A) and 24 h (B) of stimulation; (C) the basal and (D) maximal respiration, and (E) the ATP production at 6 h and 24 h after stimulation with the gp33 peptide or anti-CD3/CD8; (F) the ATP production rate; and (G) the glucose (measured using the fluorescent glucose analog 2-NBDG) and (H) the fatty acid uptake capacity (quantified using bodipy FLC$_{16}$ fluorescence intensity). *P* values from left to right (C): *P* < 1E-15, *P* = 2.67E-06. *P* values from left to right (D): *P* < 1E-15, *P* = 7.29E-06. *P* values from left to right (E): *P* < 1E-15, *P* = 0.0002. *P* values from left to right (F): *P* = 1.13E-05, *P* = 0.0003, *P* = 1.46E-05, *P* = 0.0007, *P* = 0.00018 (G). *P* values from left to right (H): *P* = 0.008, *P* = 2.82E-09, *P* = 1.06E-09. Statistical significance was determined by two-way ANOVA. (I) Representative transmission electron microscopy images of WT p14 and *Irf5*$^{-/-}$ p14 CD8 T cells stimulated with gp33 peptide or anti-CD3/anti-CD28 after 24 h (*n* = 15). (J-O) Graphs show (J) the number of mitochondria per cell, (K) the length of the mitochondria perimeter, (L) the number of *cristae* per mitochondrion, (M) the ratio of total *cristae* area per area of a mitochondrion, (N) the mean cristae width, and (O) the length of mitochondria-endoplasmic reticulum contact between WT p14 and *Irf5*$^{-/-}$ p14 CD8 T cells in the absence and presence of gp33 peptide or anti-CD3/anti-CD28. *P* = 0.0006 (J). *P* = 3.1E-05 (K). *P* = 0.036 (L). *P* values from left to right (M): *P* = 1.28E-06, *P* = 9.13E-06. *P* values from left to right (N): *P* = 14E-15, *P* = 0.002. *P* = 0.027 (O). Statistical significance was determined by *t* test. Data represent the mean ± SEM, *n* = 4, one of three independent experiments is shown, \**P* < 0.05, \*\**P* < 0.01, \*\*\**P* < 0.001, \*\*\*\**P* < 0.0001.

unable to produce ATP through both pathways (Fig. 6F). These defects were observed in *Irf5*$^{-/-}$ p14 cells only 24 h after stimulation and were more accentuated following gp33 than anti-CD3/CD28 stimulation (Fig. 6A–F). This was not due to their incapacity to uptake glucose, because *Irf5*$^{-/-}$ p14 cells could internalize the fluorescent glucose analog 2-NBDG at levels similar to the IRF-5 sufficient cells following gp33 stimulation (Fig. 6G). However, a significant difference in the capacity to uptake 2-NBDG was observed following anti-CD3/CD28 stimulation in IRF-5-deficient compared with WT p14 cells Fig. 6G). Moreover, IRF-5 deficiency did not impair the capacity to uptake lipids either. Indeed, *Irf5*$^{-/-}$ p14 cells either displayed similar capacities following anti-CD3/CD28 stimulation or were superior in internalizing fatty acids 6 h and 24 h after stimulation with the gp33 peptide, when compared with their WT counterpart (Fig. 6H). Similar results, although less dramatic, were obtained using male WT and *Irf5*$^{-/-}$ p14 cells (Appendix Fig. S8A–J). In sum, our results indicate that IRF-5-deficiency severely impairs CD8 T cell respiration and energy production.

To better understand why IRF-5-deficiency in p14 cells resulted in such a dramatic metabolic defect, we next performed electron microscopy on WT and *Irf5*$^{-/-}$ p14 T cells stimulated with gp33 or anti-CD3/CD28 (Fig. 6I). We found that *Irf5*$^{-/-}$ p14 T cells stimulated with gp33 had fewer mitochondria (Fig. 6J) but with a bigger perimeter (Fig. 6K) than their WT counterparts. Although only a small difference in the number of *cristae* per mitochondrion was observed in *Irf5*$^{-/-}$ p14 T cells following stimulation with anti-CD3/CD28 (Fig. 6L), a significant decrease in the volume of the cristae (Fig. 6M) and significant increase in the mean cristae width (Fig. 6N) were observed in IRF-5-deficient p14 cells in both stimulation conditions. Finally, we only noticed a slight but significant difference in the length of the contact site between mitochondria and endoplasmic reticulum that was shorter in IRF-5-deficient cells after anti-CD3/CD28 stimulation compared with WT p14 cells (Fig. 6O). Together, our results indicate that IRF-5-deficiency not only severely impairs CD8 T cell respiration and energy production but alters the mitochondria morphology as well.

## *Irf5*$^{-/-}$ p14 cells display impaired mitochondrial functions and heightened lipid peroxidation following LCMV Cl13 infection

To confirm that mitochondria were not functioning optimally in the absence of IRF-5 in vivo as well, we next assessed the ex vivo metabolic capacity of adoptively transferred *Irf5*$^{-/-}$ and WT p14 cells at d15 and 21 after LCMV Cl13 infection. Because adoptively transferred *Irf5*$^{-/-}$ p14 cells are present in extremely low numbers in the spleen of LCMV Cl13-infected mice at d15 and 21 p.i., we decided to use the recently described method called SCENITH (single-cell energetic metabolism by profiling translational inhibition) (Arguello et al, 2020) to evaluate the metabolic activity of our cells. This flow cytometry-based approach is based on the detection of protein synthesis levels as a proxy for the cell metabolic activity and can be used to indirectly profile the energy metabolism of rare cell populations (Arguello et al, 2020). To this end, we performed an in vivo adoptive transfer experiment as described in Appendix Fig. S1C. We found that *Irf5*$^{-/-}$ and WT p14 cells had similar translational levels at both time points analyzed (Fig. 7A), suggesting that they had a similar metabolic capacity. However, when we assessed the translational level that depended on ATP production by mitochondria, we observed a significant decrease in the capacity of *Irf5*$^{-/-}$ p14 cells to fuel translation at d21 p.i., when compared with WT p14 cells (Fig. 7B). In contrast to the results obtained from in vitro stimulations, no differences were noticed when we evaluated the glucose-dependent translational level (Fig. 7C). Interestingly, when we analyzed the mitochondria-dependency in various exhausted p14 cell subpopulations, we found that the defect in energy production by mitochondria was not present in Tex$^{\text{Prog1}}$ (Fig. 7D) and Tex$^{\text{Prog2}}$ (Fig. 7E), but only pertained to the Tex$^{\text{int}}$ (Fig. 7F) and Tex$^{\text{term}}$ (Fig. 7G) *Irf5*$^{-/-}$ p14 cells at d21 p.i. These results suggest that IRF5-deficiency mostly impacts the mitochondria capacity to produce energy in effector and terminally exhausted cells and that a defect in mitochondrial functions is also observed in vivo in *Irf5*$^{-/-}$ p14 cells.

As previously observed in in vitro stimulated cells (Fig. 6F,G), IRF-5 deficiency did not alter the cells' capacity to internalize glucose (Fig. 7H) or fatty acids (Fig. 7I).

To corroborate our results suggesting a defect in mitochondrial functions, we next measured the mitochondrial membrane potential ex vivo in cells from LCMV Cl13-infected female mice adoptively transferred with *Irf5*$^{-/-}$ and WT p14 cells. Interestingly, we noticed a significant fluorescence drop at d15 and 21 p.i. in *Irf5*$^{-/-}$ compared with WT p14 cells (Fig. 7J). We next assessed the mitochondrial mass by MitoTracker Green and found that *Irf5*$^{-/-}$ p14 cells had a larger mitochondrial mass at d21 p.i. compared with WT p14 cells (Fig. 7K), which indicates that the drop of mitochondrial membrane potential observed at d21 p.i. is even more pronounced when normalized to the mitochondrial mass

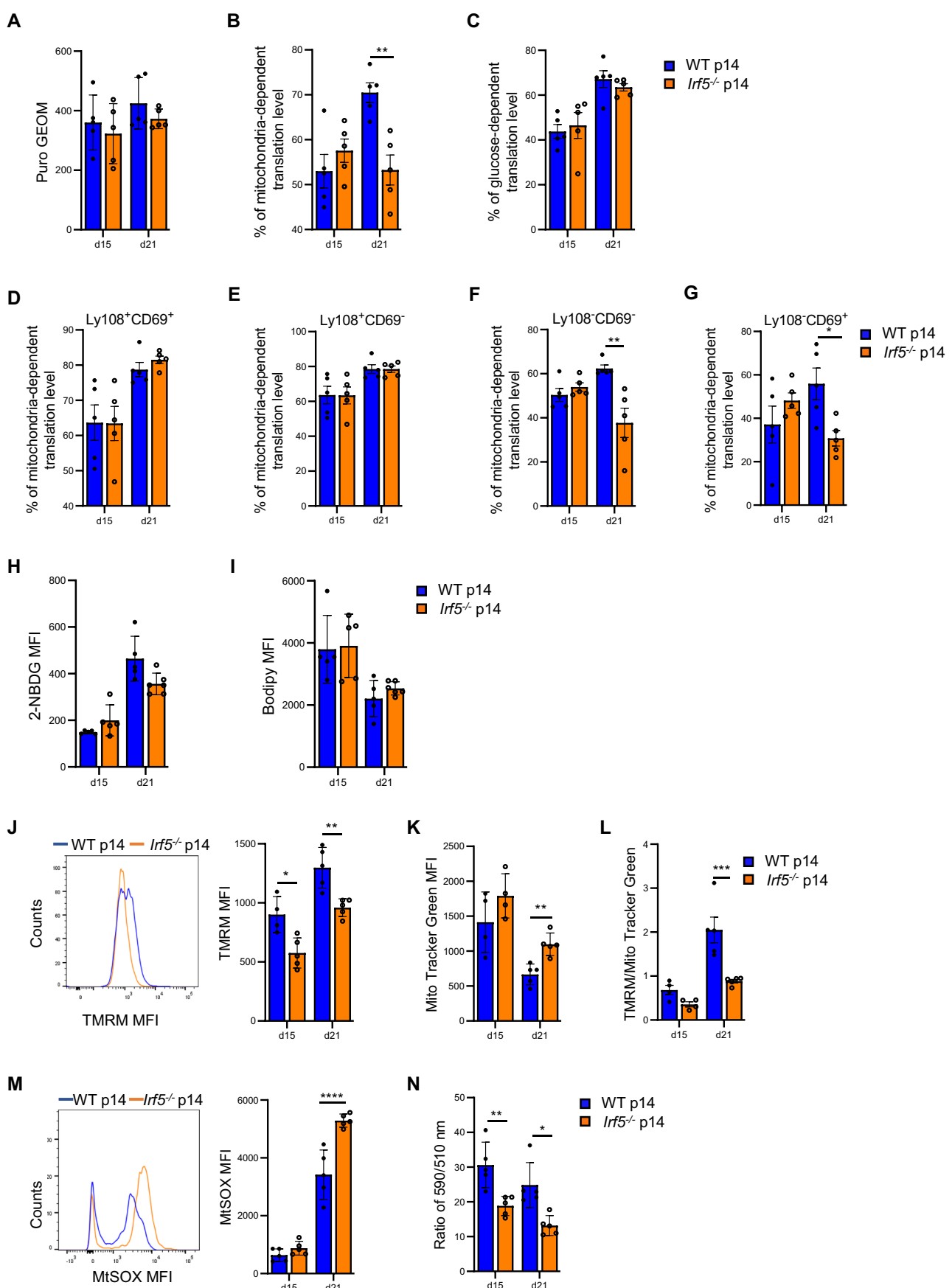

**Figure 7. _Irf5$^{-/-}$_ p14 cells display impaired mitochondrial functions and heightened lipid peroxidation.**

WT and _Irf5$^{-/-}$_ p14 CD8 T cells were adoptively transferred into female CD45.1 recipient mice a day prior to infection with LCMV Cl13. Mice were euthanized at d15 and 21 p.i. (A–G) The SCENITH kit was used to estimate the cell metabolic activity based on the protein translation levels; translation levels were assessed by determining puromycin incorporation. Graphs show (A) the total protein translation level expressed as geometrical mean of fluorescent puromycin incorporation, and the percentage decrease in protein translation following incubation with (B) oligomycin A and (C) 2-deoxy-ᴅ-glucose in adoptively transferred WT and _Irf5$^{-/-}$_ p14 CD8 T cells. (D–G) Graphs illustrate the percentage decrease in translation following incubation with oligomycin A in (D) Tex$^{Prog1}$, (E) Tex$^{Prog2}$, (F) Tex$^{int}$, and (G) Tex$^{term}$ WT and _Irf5$^{-/-}$_ p14 CD8 T cells. $P = 0.0049$ (B), $P = 0.0017$ (F), $P = 0.046$ (G). Representative graphs showing (H) the glucose uptake capacity, measured using the glucose analog 2-NBDG, (I) the bodipy FLC$_{16}$ mean fluorescence intensity as an indicator of the fatty acid uptake capacity; (J) the mitochondrial membrane potential, determined by TMRM mean fluorescence intensity; (K) the mitochondrial mass, assessed using Mito Tracker Green; (L) the membrane potential per mitochondrial mass; (M) the mitochondrial ROS production, measured by mtSOX fluorescence intensity; and (N) the relative cellular lipid peroxidation level, detected using Image-iT™ Lipid Peroxidation Kit and calculated using the ratio of the signal from the 590 nm and 510 nm channels (lower ratios indicate higher lipid peroxidation levels) in WT and _Irf5$^{-/-}$_ p14 CD8. $P$ values from left to right (J): $P = 0.013$, $P = 0.0063$. $P = 0.0059$ (K). $P = 0.00097$ (L). $P = 5.79E-05$ (M). $P$ values from left to right (N): $P = 0.0095$, $P = 0.01$. Statistical significance was determined by two-way ANOVA. Data represent the mean ± SEM, n = 5, *$P < 0.05$, **$P < 0.01$, ***$P < 0.001$, ****$P < 0.0001$.

(Fig. 7L). Moreover, we observed significant difference in mitochondrial reactive oxygen species (ROS) levels ex vivo in _Irf5$^{-/-}$_ compared with WT p14 cells at d21 p.i. (Fig. 7M). Hence, we reasoned that if _Irf5$^{-/-}$_ p14 cells can internalize lipids equally as well as WT p14 cells and ROS levels are elevated, this means that perhaps lipid peroxidation occurs in those cells. To this end, we measured lipid peroxidation ex vivo and found, indeed, that lipid peroxidation was occurring in IRF-5-deficient p14 cells at d15 and d21 p.i. (Fig. 7N). In summary, our results demonstrate that in the absence of IRF-5, CD8 T cells enter the cell cycle more rapidly, but are metabolically stressed because of a severe defect in mitochondrial functions, become more rapidly exhausted, present increased lipid peroxidation, and finally die.

## Discussion

In the past decade, several studies have uncovered numerous transcription factors governing the intricate gene expression program underlying CD8 T cell exhaustion during chronic infections. However, relatively little is known about factors that help CD8 T cell to survive in the hostile chronic inflammatory environment typical for persistent infections. Here, we show that IRF-5 prevents premature functional exhaustion and cell death by controlling the cell cycle and acting like a metabolic checkpoint in CD8 T cells during the chronic stage of infection. Indeed, IRF-5-deficient CD8 T cells quickly become functionally exhausted, display severe defects in mitochondrial structure and functions, and ultimately die.

IRF-5 function has been studied in detail in antigen presenting cells and tumor cells, where it regulates the inflammatory and anti-viral response (Barnes et al, 2004), macrophage polarization (Krausgruber et al, 2011), cell cycle (Barnes et al, 2003), and cell death (Hu and Barnes, 2009; Hu et al, 2005). Recently, a function for IRF-5 in governing the carbohydrate metabolism was also ascribed (Albers et al, 2021; Hedl et al, 2016). In contrast, the literature on the role of IRF-5 in T cells is sparse and only describes IRF-5 functions in CD4 T cells (Fabie et al, 2018; Yan et al, 2020). We have now shown that IRF-5's role substantially differs between CD4 and CD8 T cells. Other transcription factors also have opposing functions in both T cell subsets. For instance, TOX plays an essential role in the CD4 T cell development (Aliahmad and Kaye, 2008), but drives exhaustion in CD8 T cells (Alfei et al, 2019; Khan et al, 2019; Scott et al, 2019; Seo et al, 2019; Yao et al, 2019).

In our infection model, the absence of IRF-5 profoundly affected CD8 T cell responses to an extent that they are incapable of clearing LCMV Cl13 infection in the serum. Unsurprisingly, we found a dysregulated cell cycle in IRF-5-deficient CD8 T cells. Indeed, IRF-5 was shown to exert cell cycle arrest functions in a p53-dependent (Mori et al, 2002) and -independent (Hu and Barnes, 2009) way in tumor cells. In LCMV Cl13-infected mice, CD8 T cells were entering the cell cycle more rapidly in the absence of IRF-5; however, IRF-5 ablation did not affect the clonal expansion, as similar numbers and frequencies of CD8 T cells were observed at d8 p.i. in WT and IRF-5 deficient cells. Moreover, _Irf5$^{-/-}$_ p14 CD8 T cells were dying more rapidly than their WT counterparts.

In the search for reasons for a premature cell death, we found that _Irf5$^{-/-}$_ p14 CD8 T cells were not capable to cope metabolically and gain energy through the lipid metabolism. Only few studies, so far, have shown a possible link between IRF-5 and the lipid metabolism. The first one by Montilla et al suggests that IRF-5 may be involved in the lipid metabolism, because _Irf5$^{-/-}$_ mice do not properly process myelin-derived lipids in a model for experimental autoimmune encephalomyelitis (EAE) (Montilla et al, 2025). Another study by Orliaguet et al demonstrates that IRF-5-deficient adipose tissue macrophages have a highly oxidative nature in the context of a short-term high fat diet. This was caused by transcriptional de-repression of the mitochondrial matrix component _Ghitm_, which is a direct target of IRF-5 (Orliaguet et al, 2022). GHITM is an inner membrane protein that maintains mitochondrial architecture for efficient oxidative phosphorylation (Oka et al, 2008) and de-repression of _Ghitm_ in _Irf5$^{-/-}$_ macrophages results in an increased oxygen consumption rate (Orliaguet et al, 2022). In our model, IRF-5-deficient p14 CD8 T cell also upregulate _Ghitm_, but do not display a higher oxygen consumption rate upon in vitro stimulation with the gp33 peptide or with anti-CD3/CD28, or ex vivo. On the contrary, the cellular respiration was dramatically impaired and _Irf5$^{-/-}$_ CD8 T cells were less capable of gaining energy from the lipid metabolism. We also found several down-regulated genes in _Irf5$^{-/-}$_ CD8 T cells that are involved in lipid synthesis (e.g., _Awat2_, _Gpat2_), electron transport chain (e.g., _Coq4_) or in the assembly of the mitochondrial respiratory chain complex I (e.g., _Ndufaf3_, _Ndufb4_). Remarkably, all downregulated genes are directly or indirectly involved in the mitochondrial oxidative phosphorylation process or are components of the mitochondrial envelop. Moreover, some of the genes that were downregulated in the absence of IRF-5, like _Ndufaf3_ and _Coq4_, are associated with mitochondrial disorders in humans (Brea-Calvo et al, 2015; Saada

et al, 2009; van der Ven et al, 2023) and with the Leigh syndrome (Baertling et al, 2017).

In agreement with our results, (Albers et al, 2021) also reported a significant decrease in oxygen consumption rates in IRF-5-deficient alveolar macrophages stimulated with TLR3, when compared with IRF-5 sufficient cells (43). Thus, it is possible that IRF-5 functions depend on the cell type, the environment, or on additional stimuli the cell may receive. Indeed, the absence of IRF-5 only affected effector Tex$^{int}$ and terminally exhausted CD8 T cells but did not alter the metabolic activity of progenitor-like Tex CD8 T cells. In our model, the membrane potential of *Irf5$^{-/-}$* p14 CD8 T cells was significantly reduced when compared with WT p14 CD8 T cells at d15 p.i., suggesting that mitochondria are defective in murine IRF-5 deficient cells. Further investigations are warranted to identify upstream signals and downstream targets of IRF-5 and understand the relationship between IRF-5 and PD-1 and Blimp-1, two molecules involved in metabolic rewiring and mitochondrial restructuring in exhausted cells (Patsoukis et al, 2015; Yu et al, 2015; McLane et al, 2021); to note, Blimp-1 is a known target of IRF-5 (Lien et al, 2010).

IRF-5 deficiency in CD8 T cells did not affect energy production via glycolysis in vivo. However, we cannot exclude that Tex$^{int}$ and terminally exhausted cells may compensate the energy deficit given by a defective oxidative phosphorylation by enhancing glycolysis. This compensatory effect could push them down the exhaustion spiral and cause premature functional exhaustion. Hedl et al reported a role for IRF-5 in enhancing glycolysis in macrophages upon pattern recognition receptor (PRR) stimulation (Hedl et al, 2016). IRF-5 was also shown to regulate glycolysis upon TLR3 stimulation of airway macrophages by directly controlling *Hk2* expression (Albers et al, 2021). This heightened expression of glycolytic genes could represent a compensatory mechanism to cope with defective lipid metabolism or derive from a skewed balance between IRF-4 and IRF-5. Although we did not see any difference in the level of IRF-4 expression (mRNA and protein) at any time point after LCMV Cl13 infection, these two transcription factors are known to compete with each other (Negishi et al, 2005). IRF-4 was also shown to regulate glycolysis in CD8 T cells and to promote expansion and differentiation into effector cells (Man et al, 2013); not surprisingly, it also promotes exhaustion (Man et al, 2017). IRF-4 is downstream of TCR signaling and links antigen affinity with the cellular metabolism. Indeed, high-affinity peptides induce glycolysis and promote differentiation into effector cells via IRF-4, which directly targets *Hk2* (encoding for Hexokinase 2; also an IRF-5 target (Albers et al, 2021)) and *Glut3* (encoding for a glucose transporter) (Man et al, 2013). Thus, it is possible that the higher frequency of KLRG1$^+$ IRF-5-deficient p14 cells compared to WT p14 CD8 T cells observed at d8 p.i. derives not only from heightened proliferation, but also from a skewed IRF-4/IRF-5 balance, where IRF-4 is left without a competitor. IRF-4 was also shown to regulate *Hif1a* (Man et al, 2013). The Hypoxia Inducible Factor-1α (HIF-1α) itself also promotes glucose metabolism and the expression of glycolytic genes in CD8 T cells (Finlay et al, 2012). Hence, it is possible that HIF-1α and IRF-4 work together to antagonize IRF-5 during clonal expansion promoting glycolysis and limiting the cell division brake imposed by IRF-5. This would be in agreement with a recent study that demonstrates how HIF-1α directly suppresses IRF-5 in monocytes (Peng et al, 2021). Like IRF-

4, hypoxia also promotes functional exhaustion (Scharping et al, 2021). Nevertheless, a skewed IRF-4/RF-5 balance cannot entirely explain the severe defects we observed in IRF-5-deficient CD8 T cells during the chronic phase of infection. Interestingly, the absence of IRF-5 mostly impacted CD8 T cell responses during the chronic phase of infection, when IRF-5 seems to act as a metabolic checkpoint and help CD8 T cells cope with the hostile environment created by the chronic inflammation and the continuous antigen stimulation. Our data also suggest an involvement of IRF-5 in regulating the mitochondria morphology. Mitochondria remodelling is essential for the proper functioning of cells. Indeed, dysregulation of mitochondrial dynamics could lead to cell death. In our study, IRF-5-deficiency was associated with several changes in the mitochondria structure: *Irf5$^{-/-}$* p14 CD8 T cells had fewer but larger mitochondria, and the mitochondrial cristae had a smaller volume density and a higher mean width compared with IRF-5-sufficient cells. Larger and fewer mitochondria could be indicative of increased mitochondria fusion following higher energy demand or stress, whereas wider cristae with smaller volume density were shown to be associated with a decreased mitochondrial respiratory efficiency (Cogliati et al, 2013). A study by Cogliati et al, describes the protein optic atrophy 1 (OPA-1) as the main regulator of mitochondrial cristae organization and clearly demonstrate that the cristae structure determines the respiratory chain supercomplexes' assembly and, ultimately, the efficiency of the mitochondria respiratory function (Cogliati et al, 2013). However, the role of IRF-5 in shaping the mitochondrial morphology in a chronic inflammatory environment and its relationship with OPA-1 still need to be determined.

Mitochondrial disfunction has recently been proposed to drive the transition from Tex$^{pr}$ to terminally exhausted cells by generating high levels of ROS and reprogramming cells towards glycolysis and exhaustion (Scharping et al, 2021; Wu et al, 2023). Interestingly, metabolic reprogramming is induced by ROS-dependent HIF-1α triggering (Wu et al, 2023). Hence, IRF-5 could be protecting cells from transitioning towards exhaustion by supporting mitochondria remodelling and promoting oxidative phosphorylation, thus acting as a metabolic checkpoint. Moreover, IRF-5 could also be involved in protecting CD8 T cells against oxidative stress at chronic stage of LCMV Cl13 infection, since *Irf5$^{-/-}$* CD8 T cells present with increased lipid peroxidation.

IRF-5 deficiency in CD8 T cells lead to premature cell death. In these cells, cell death could have occurred because of exhaustion, the severe bioenergetic defects, or as a consequence of lipid peroxidation. Lipid peroxidation is a well-known cause of cell death and typically leads to membrane rupture (Gaschler and Stockwell, 2017). Moreover, lipid peroxidation can induce apoptosis (Kagan et al, 2005), necroptosis (Canli et al, 2016), pyroptosis (Kang et al, 2018), or be the primary driver of ferroptosis (Dixon et al, 2012). Future studies will investigate the cause of death in IRF-5-deficient CD8 T cells and the possible role of IRF-5 in limiting oxidative stress.

In conclusion, we have identified a role for IRF-5 in protecting CD8 T cells from premature functional exhaustion and cell death by acting as a metabolic checkpoint during chronic LCMV Cl13 infection, supporting mitochondrial remodelling and oxidative phosphorylation. These results highlight the central role of mitochondrial function in governing CD8 T cell exhaustion.

# Methods

## Reagents and tools table

| Reagent/resource | Reference or source | Identifier or catalog number |
| --- | --- | --- |
| **Experimental models** | | |
| *Irf5^flox/flox* mice | Jackson Laboratories | Strain #017311 |
| p14 mice | Gift from Dr. Alain Lamarre | NA |
| **Recombinant DNA** | | |
| **Antibodies** | | |
| Syrian hamster anti-CD3 | eBioscience | Clone 500A2 |
| Rat anti-CD8 | BD Bioscience | Clone 53-6.7 |
| Rat anti-CD44 | BD Bioscience | Clone IM7 |
| Rat anti-CD62L | BD Bioscience | Clone MEL-14 |
| LIVE/DEAD™ Fixable Near IR (780) Viability Kit | ThermoFisher | L34994 |
| Mouse anti-CD45.2 | BD Bioscience | Clone 104 |
| Annexin V-APC | BD Bioscience | Catalog#:550475 |
| Hamster anti-KLRG1 | eBioscience | Clone 2F1 |
| Rat anti-CD127 | eBioscience | Clone SB/199 |
| Rat anti-LAG-3 | Biolegend | Clone C9B7W |
| Rat anti-PD-1 | Biolegend | Clone 29F.1A12 |
| Rat anti-TIM-3 | Biolegend | Clone RMT3-23 |
| Mouse anti-CX3CR1 | eBioscience | Clone SA011F11 |
| Rat anti-CD101 | Biolegend | Clone Moushi101 |
| Mouse anti-Ly108 | eBioscience | Clone 330-AJ |
| Armenian Hamster anti-CD69 | BD Bioscience | Clone H1.2F3 |
| streptavidin-V500 | | |
| Mouse anti-Tbet | eBioscience | Clone 4B10 |
| Rat anti-EOMES | eBioscience | Clone Dan11mag |
| Rat anti-IRF4 | eBioscience | Clone 3E4 |
| Human cell line anti-TOX | Milteniy Biotech | Clone REA473 |
| Rat anti-Ki67 | eBioscience | Clone SolA15 |
| Rat anti-IRF5 | Biolegend | W16007B |
| Rat anti-IFNγ | BD Biosciences | XMG1.2 |
| Rat anti-TNF | BD Biosciences | MP6-XT22 |
| Rat anti-IL2 | eBioscience | JES6-5H4 |
| PE-gp$_{33-41}$ tetrameric complexes | synthesized in house | NA |
| **Oligonucleotides and other sequence-based reagents** | **Forward** | **Reverse** |
| Awat2 | GGAGAGACAGAC | TCAAACTATCACCAGCTC |
| Gpld1 | CTCTATGACCAGC | CTGGGTC |
| Coq4 | TCTTGGCAACAGA | CCAGTCAGCTTCCTCCAA |
| Gpat2 | TGCAGAC | AG |
| Ndufb4 | ACGAGCACCTAC | GCACAGCCTCAAACCACT |
| Ndufaf3 | ACGCTTTGTG | TTACC |
| Ghitm | GCCATTGCCTATG | CAGTCTCCGAAAGACAG |
| mGAPDH | ACCTGGTTC | CCAAG |
| Actin | GCCGAGTATGAC | GCATAGGTCCAGCGAATC |
| | GTGTCTCCG | AAGG |
| | GCTACAGTAGCA | CAGCATCCAGAAAAGGGA |
| | GAGGCTTCAC | GAAGC |
| | CTGCATTCTGGTG | TGAGTACAGAGTGGCACC |
| | TGATGGG | AG |
| | AGGTCGGTGTGA | TGTAGACCATGTAGTTGA |
| | ACGGATTTG | GGTCA |
| | GGCTGTATTCCC | CCAGTTGGTAACAATGCC |
| | CTCCATCG | ATGT |

| Reagent/resource | Reference or source | Identifier or catalog number |
| --- | --- | --- |
| **Chemicals, enzymes and other reagents** | | |
| gp$_{33-41}$: KAVYNFATC (LCMV-GP, H-2Db) | New England Peptide (now Biosynth; Gardner, MA, USA) | Clone 16-0031-85 Clone 16-0281-85 |
| Trizol | Sigma | N13195 |
| recombinant human IL-2 | Peprotech | D3821 |
| fetal bovine serum | Wisent Bioproduct | M46750 |
| Brefeldin A | BD Bioscience | M36008 |
| RNeasy mini kit | Qiagen | T668 |
| KAPA RNA HyperPrep | Roche | C10445 |
| anti-mouse CD3 | eBioscience | |
| anti-mouse CD28 | eBioscience | |
| 2-NBDG | ThermoFisher Scientific | |
| Bodipy | ThermoFisher Scientific | |
| SCENITH kit | https://www.scenith.com | |
| Mitotracker Green | ThermoFisher | |
| MitoSOX Red | ThermoFisher | |
| TMRM | ThermoFisher | |
| Image-iT® Lipid Peroxidation Sensor | ThermoFisher | |
| **Software** | | |
| Flowjo v10.8.1 | | |
| Trimmomatic version 0.39 | Agilent | |
| HISAT2 version 2.2.1 | Bio-Rad | |
| featureCounts (Subread package version 2.0.6) | | |
| GRCm38-Version M10 (Ensembl 85) | | |
| EdgeR package version 4.2.1 | | |
| Wave software | | |
| GraphPad Prism 8.0 | | |
| ImageJ | | |
| QX manager software 2.0 | | |
| **Other** | | |
| BD LSRFortessa II | BD LSRFortessa II | |
| ImageStreamX | Cytek Biosciences | |
| Illumina NextSeq500 | Illumina | |
| XF-96 Extracellular Flux Analyzer | Agilent | |
| Hitachi H-7100 TEM | Hitachi | |
| QX200 Droplet Digital PCR system | Bio-Rad | |
| BD FACSAria IIu | BD Biosciences | |

## Mice

C56BL/6-CD45.1 and *Irf5^flox/flox* mice were purchased from The Jackson Laboratory. *Irf5^−/−* mice were generated by breeding *Irf5^flox/flox* mice with mice expressing *Cre*-recombinase under the CMV promoter. Because of the CMV promoter, IRF-5 ablation in these mice is ubiquitous. Mice with a targeted IRF-5 mutation in T cells were generated by breeding *Irf5^flox/flox* mice with mice expressing

the *Cre*-recombinase under the Lck promoter (Fabie et al, 2018). LCMV gp33 antigen-specific p14 mice were crossed with *Irf5flox/flox* mice to obtain *Irf5 flox/flox* p14 mice. These mice were crossed with *Irf5−/−* mice to obtain IRF-5 deficient LCMV gp33-antigen-specific p14 mice (*Irf5−/−* p14). The glycoprotein GP$_{33-41}$-specific TCR transgenic P14 mice were kindly provided by A. Lamarre. All mice were housed and bred at the INRS animal facility under specific pathogen-free conditions and were used at 6–10 weeks of age.

## Ethical statement

In vivo experiments were performed under protocols approved by the Comité Institutionnel de Protection des Animaux of the INRS - Centre Armand-Frappier (#1910-01, #2003-02). These protocols respect procedure on good animal practice provided by the Canadian Council on Animal Care.

## Viral production and titration

LCMV clone 13 was originally obtained from Dr. Sam Basta (Queens University) and expanded on L-929 cells in EMEM (Wisent) supplemented with 1 mM sodium pyruvate (Wisent) and 5% FCS (PAA). Cells were infected at a MOI of 0.02 and supernatants collected and cleared of debris 48 h after infection. Viral titers of viral stocks and processed organs were determined on MC57 cells using a standard LCMV focus-forming assay (Battegay et al, 1991).

## Peptide and tetramers

The synthetic peptide gp$_{33-41}$: KAVYNFATC (LCMV-GP, H-2Db) was purchased from New England Peptide (now Biosynth; Gardner, MA, USA). PE-gp$_{33-41}$ tetrameric complexes were synthesized in-house and used to detect LCMV-specific CD8 T cells (Altman et al, 1996). Briefly, splenocytes were first stained with PE-gp$_{33-41}$ tetramers for 30 min at 37 °C, followed by direct surface staining with the described antibodies for another 20 min on ice (Lacasse et al, 2008).

## Cell isolation and adoptive transfer experiments

LCMV-specific CD8 p14 T cells were isolated using negative selection (Miltenyi Biotec) from spleens of *Irf5flox/flox*–*CMV-Creneg*-p14 (WT p14) and *Irf5flox/flox*–*CMV-Cre+*-p14 (*Irf5−/−* p14) mice as previously described (Hammami et al, 2015). For adoptive transfer experiments, cells were stained with anti-CD3-Alexa fluor 700 (500A2, eBioscience), anti-CD8-PE (53-6.7, BD Bioscience), anti CD44-PE-Cy7 (IM7, BD Bioscience), anti CD62L-APC (MEL-14, BD Bioscience), and naive cells were sorted based on their CD62L$^{high}$ CD44 $^{low}$ phenotype. 2000 cells were transferred intravenously into sex-matched congenic CD45.1 mice 1-day prior infection with $2 \times 10^6$ PFU LCMV clone 13.

For RNA sequencing and digital droplet PCR (ddPCR), adoptively transferred p14 cells was sorted 21 days post infection. For RNA sequencing experiment, 4 samples of 4 individual experiments including 2 males and 2 females per group were used. 8 WT males and 10 KO males were pooled for the 1st experiment, 8 WT males and 17 KO males for the 2nd experiment, 7 WT females and 17 KO females for the 3rd experiment, and 8 WT females and

17 KO females for the 4th experiment. For ddPCR experiment, 3 samples of 3 experiments using females were used. Each sample is the pool of isolated cells from 10 to 15 spleens. Cells were stained with anti-CD3-Alexa Fluor 700 (500A2, eBioscience), anti-CD8-PE (53-6.7, BD Bioscience), anti-CD45.2- FITC (104, BD Bioscience). Cells were sorted on a BD FACSAria IIu cell sorter from BD Biosciences, 3 laser configurations (Blue 488 nm, Red 633 nm, Violet 405 nm), nozzle size 70-micron, sheath pressure 70 psi. Cell sorting was performed at the Flow cytometry platform of the Institute for Research in Immunology and Cancer's Genomics (Université de Montreal). CD8$^+$ CD45.2$^+$ population was collected directly into 500 µl Trizol (Sigma) and stored at −80 °C until used.

## Flow cytometry

Cells were washed with PBS and stained with LIVE/DEAD™ Fixable Near IR (780) Viability Kit (L34994, ThermoFisher) for 5 min at room temperature. After washing with FACS buffer, surface and intracellular staining was performed as previously described (Hammami et al, 2015; Joshi et al, 2009). The following fluorochrome-conjugated antibodies were used for surface staining anti-CD3-Alexa fluor 700 (500A2, eBioscience), anti-CD8-PB, aniti CD8-PE, anti-CD8-APC (53-6.7, BD Bioscience), anti-CD45.2-FITC, anti-CD45.2-PE, anti-CD45.2- Alexa flour 700 (104, BD Bioscience), Annexin V APC (BD Biosciences), anti KLRG1-APC (2F1, eBioscience), anti CD127-PE (SB/199, BD Bioscience), anti-LAG-3-PE-Cy7 (C9B7W, eBioscience), anti-PD-1-biotin (29F.1A12, Biolegend), anti-TIM-3-BV605 (RMT3-23, Biolegend), anti CD44-PE-Cy7 (IM7, BD Biosciences), anti CX3CR1-PerCP-Cy5.5 (SA011F11, Biolegend), anti CD101-Alexa flour 700 (Moushi101, eBioscience), anti-Ly108-APC, anti-Ly108-PE (330-AJ, Biolegend), anti-CD69-PerCP Cyanine 5.5 (H1.2F3, eBioscience), and streptavidin-V500 (BD Bioscience). Cells were fixed using 2% paraformaldehyde and permeabilized using 0.1% saponin solution. Cells were stained with cleaved Caspase-3 (Asp175) PE (clone D3E9, Cell Signaling Technology) to measure cell apoptosis. Expression of T-bet, EOMOES, IRF-4, and TOX was measured by staining cells with anti-Tbet-PerCP-Cy5.5 (4B10, eBioscience), anti-EOMES-PE (Dan11mag, eBioscience), anti-IRF4-PECy-7 (3E4, eBioscience), and anti-TOX-PE (REA473, Miltenyi). For cell cycle analysis, cells were stained using anti-Ki67-PE-Cy7 (SolA15, eBioscience), washed with PBS, and labelled with DAPI (10 µg/ml final concentration) at room temperature for 10 min. To measure IRF-5 expression, permeabilized cells were stained with anti-IRF5-PE (W16007B, Biolegend) and analyzed by flow cytometry or then stained with DAPI and acquired by ImageStreamX as previously described (Fabie et al, 2018).

For cytokine quantification, splenocytes were restimulated for 5 h at 37 °C with 1 µg/ml of MHC class I-restricted LCMV peptide gp$_{33-41}$ in the presence of 2 ng/ml recombinant human IL-2 (Peprotech), 10% fetal bovine serum (premium FBS, Wisent Bioproduct), and 1 µg/ml Brefeldin A (BD Bioscience). Following ex vivo restimulation, cells were stained with stained, fixed, and permeabilized as described above. The following antibodies were used for intracellular staining: anti-IFNγ-APC (XMG1.2, BD Biosciences), anti-TNF-PE-Cy7 (MP6-XT22, BD Biosciences), and anti-IL2-PE (JES6-5H4, eBioscience). Samples were acquired on a BD LSRFortessa II cell analyzer (Becton Dickinson). Data was analyzed with FlowJo v10.8.1.

## Bulk RNA sequencing

Total RNA was isolated using RNeasy mini kit (Qiagen) according to the manufacturer's instructions. RNA was quantified using Qubit (Thermo Scientific) and quality was assessed with the 2100 Bioanalyzer (Agilent Technologies). Transcriptome libraries were generated using the KAPA RNA HyperPrep (Roche) using a poly-A selection (Thermo Scientific). Sequencing was performed on the Illumina NextSeq500, obtaining around 20 M reads per sample. Bulk RNA sequencing was performed at the Genomics Platform of the Institute for Research in Immunology and Cancer (Université de Montreal).

## RNA seq analysis

Adapter sequences and low-quality bases in the resulting FASTQ files were trimmed from raw sequences using Trimmomatic version 0.39 (Bolger et al, 2014) with the following arguments: SE ILLUMINACLIP:TruSeq-SE_adapters.fa:2:30:10 LEADING:3 TRAILING:3 SLIDINGWINDOW:4:15 MINLEN:36. Genome alignment was conducted for trimmed reads using HISAT2 version 2.2.1 (Kim et al, 2019) on the mm10 mouse reference genome assembly. Expression levels were quantified using strand-aware counting of reads aligning to exonic features using featureCounts (Subread package version 2.0.6) (Liao et al, 2014) with annotations from GRCm38-Version M10 (Ensembl 85). Raw read counts were filtered to remove residual rRNA and mitochondrial reads, and only genes with an expression level above a threshold of 10 counts per million reads (CPM) in at least 6 libraries were kept, yielding a total of 8990 expressed genes. A principal component analysis (PCA) was computed on the CPM values. Filtered library CPMs were normalized using TMM normalization before then being fit to a genewise negative binomial generalized linear model using the EdgeR package version 4.2.1 (Robinson et al, 2010). The differential gene expression analysis evaluated various contrasts using genewise likelihood ratio tests, to test the impact of the infection and genotype. Significantly dysregulated genes were selected using an absolution log2 fold change threshold of >1 and an FDR < 0.05 (Benjamini–Hochberg corrected $P$ values).

## Metabolic assays

CD8 T cells from WT p14 or $Irf5^{-/-}$ p14 mice were isolated as described above and cultured in complete RPMI medium supplemented with 2 ng/ml recombinant human IL-2 (Peprotech) alone, with 1 μg/ml gp33 peptide, or with 1 μg/ml anti-mouse CD3 (eBioscience, 16-0031-85) and 5 μg/ml anti-mouse CD28 (eBioscience, 16-0281-85). Cells were then collected at 6 or 24 h after incubation and washed with PBS. For mito stress and real-time ATP production experiments, 4 replicates per condition were used and $2 \times 10^5$ cells were plated for each replicate. Cells were resuspended in Agilent RPMI media containing 10 mM glucose, 2 mM glutamine, and 1 mM sodium pyruvate and incubated at 37 °C for 1 h without $CO_2$. Mitochondria respiration, oxygen consumption rates (OCR), and extracellular acidification rates (ECAR) were measured using the XF-96 Extracellular Flux Analyzer (Agilent) at baseline and in response to 1.5 μM Oligomycin, 2 μM FCCP, and 0.5 μM rotenone/antimycin A, according to manufacturer's protocols (Mito Stress Test Kit,

Agilent). For real-time ATP production, oxygen consumption rates (OCR) and extracellular acidification rates (ECAR) were measured at baseline and in response to 1.5 μM Oligomycin and 0.5 μM rotenone/antimycin A following manufacturer's protocols. The results were analyzed using Wave software (Agilent).

To measure glucose and lipid uptake capacity, cells were labelled with 150 μM of the fluorescently labeled glucose analog 2-NBDG (ThermoFisher Scientific, N13195) or with 1 μM bodipy (Thermo-Fisher Scientific, D3821) in serum-free media for 30 min at 37 °C. Cells were washed with PBS prior to cell surface staining, as described above, and analyzed by flow cytometry.

## Transmission electron microscopy

CD8 T cells from WT p14 or $Irf5^{-/-}$ p14 mice were isolated as described above and cultured in complete RPMI medium supplemented with 2 ng/ml recombinant human IL-2 (Peprotech) alone, with 1 μg/ml gp33 peptide, or with 1 μg/ml anti-mouse CD3 (eBioscience, 16-0031-85) and 5 μg/ml anti-mouse CD28 (eBioscience, 16-0281-85). Cells were then collected 24 h after incubation and washed with PBS. Cells were fixed with 2.5% glutaraldehyde in 0.05 M cacodylate buffer (EMS), pH 7.4, over-night at 4 °C, rinsed in cacodylate buffer three times for five minutes each, and then incubated with freshly prepared 1.3% (w/v) osmium tetroxide in collidine buffer for 1 h. Cells were then dehydrated by successive passage through 25, 50, 75, and 95% solutions of acetone (Fischer) in water for 30 min each, followed by immersion in two changes of pure acetone for at least 30 min each. Subsequently, cells were immersed for 16–18 h in SPURR (TedPella): acetone (1:1), embedded by immersion in two successive baths of SPURR (TedPella), for at least 2 h each, cut into small pieces, and placed in BEEM capsules (TedPella). The capsules were then filled with SPURR mixtures and let stand at room temperature for 18 h. The filled capsules were then placed at 60 °C for 24 h to allow the resin to polymerize, cut into ultra-thin sections (90 nm thick) on an ultramicrotome (Leica UC7), put onto 200 mesh copper grids covered with Formvar and carbon, and finally stained with uranyl acetate 5% in 50% ethanol for 15 min, followed by lead citrate for 5 min. Electron micrographs were produced using a Hitachi H-7100 TEM with AMT XR111 camera in different magnifications. The number and morphology of mitochondria were analyzed using ImageJ.

## Single-cell energetic metabolism by profiling translation inhibition (SCENITH)

SCENITH was performed as described in Arguëllo et al (Arguello et al, 2020) using the SCENITH kit (https://www.scenith.com/). Splenocytes were washed with 1× PBS before the treatment during 20 min with Control, metabolic inhibitors 2-deoxy-D-glucose (DG, final concentration 100 mM) or oligomycin A (Oligo, final concentration 1 μM), or with harringtonine (final concentration 2 μg/mL) as negative control. Without washing, puromycin (final concentration 10 μg/mL) is added and cells were incubated for further 15 min. After puromycin incubation, cells were washed in cold 1× PBS and stained with LIVE/DEAD™ Fixable Near IR (780) Viability Kit (L34994, ThermoFisher), then with primary con-jugated antibodies against cell surface markers, and were finally fixed and permeabilized as described above. Cells were

intracellularly stained with anti-puro-Alexa Fluor 647 and acquired on a BD LSRFortessa II cell analyzer (Becton Dickinson).

Mitochondria-dependent and glycolysis-dependent translation levels were calculated as percentage decrease in translation level following oligomycin A or 2-deoxy-D-glucose treatments, respectively, using the following formulas:

$$\% \text{ of mitochondria-dependent translation level} = (Co - O)/Co \times 100$$

$$\% \text{ of glycolysis-dependent translation level} = (Co - DG)/Co \times 100$$

Whereby, Co = GeoMFI of anti-puro-fluorochrome upon Control treatment; O = GeoMFI of anti-puro-fluorochrome upon oligomycin A treatment; and DG = GeoMFI of anti-puro-Fluorochrome upon 2-deoxy-D-glucose treatment.

## Mitotracker green, MitoSox and tetramethylrhodamine methyl ester (TMRM)

Mitotracker Green (ThermoFisher, M46750) was used to measure mitochondria mass following the manufacturer's instructions. Briefly, cells were washed with PBS, then added 1:1000 reconstitute Mitotracker Green and were incubated 30 min at 37 °C. To measure mitochondrial superoxide and the mitochondrial membrane potential state, cells were labelled with 1 μM MitoSOX Red mitochondrial superoxide indicator reagent (ThermoFisher, M36008) or with 100 nM TMRM (ThermoFisher, T668) and were incubated for 30 min at 37 °C. Cells then were washed with PBS prior to cell surface staining as described above and analyzed on a BD LSRFortessa II cell analyzer (Becton Dickinson).

## Lipid peroxidation

To evaluate lipid peroxidation, cells were labelled with Image-iT® Lipid Peroxidation Sensor (ThermoFisher, C10445) at a final concentration of 5 μM, then were incubated for another 30 min at 37 °C. Cells were then washed three times with PBS and stained with other cell surface antibodies. The fluorescence signals were read at separate wavelengths by flow cytometry; one at 581/591 nm for the live dye, and the other at 488/510 nm for the oxidized dye. The ratio of the emission fluorescence intensities at 590 nm to 510 nm gives the read-out for lipid peroxidation in cells. Lower ratio corresponds to higher level of lipid peroxidation.

## Digital droplet PCR

Total RNA was isolated using RNeasy mini kit (Qiagen) according to the manufacturer's instructions. Reverse transcription was performed using the iScript cDNA synthesis kit (Bio-Rad) following the manufacturer's protocol. In total, 25 μl reaction mix for ddPCR was made by 12.5 μl EvaGreen supermix, 250 nM forward and reverse primers, 2 μl cDNA and water. ddPCR was performed using a QX200 Droplet Digital PCR system (Bio-Rad). Specifically, 20 μl of each reaction mix and 70 μl of oil were converted to droplets with the QX200 droplet generator (Bio-Rad). Droplet-partitioned samples were then transferred to a 96-well plate, sealed, and cycled in a C1000 Touch Thermocycler (Bio-Rad) under the cycling protocol that was optimized for each primer. The

cycled plate was then transferred and read in the QX200 reader (Bio-Rad) and analyzed by QX Manager software 2.0 (Bio-Rad). Fold change expression relative to naive was calculated by normalized with *Gapdh* and *Actb*.

## Statistical analysis

GraphPad Prism 8.0 was used for statistical analysis. Each value represents at least three independent experiments. Two-tailed Student *t* test or ordinary one-way ANOVA or two-way ANOVA were used to determine statistical significance. $*P < 0.05$, $**P < 0.01$, $***P < 0.001$, $****P < 0.0001$. Error bars denote ± SEM.

## Data availability

The RNAseq data from this publication have been deposited to the GEO database (https://www.ncbi.nlm.nih.gov/geo/) and assigned the accession number GSE292794. Data used in this study have been deposited to the BioStudies database and were assigned the accession number S-BSST1936.

The source data of this paper are collected in the following database record: biostudies:S-SCDT-10_1038-S44318-025-00485-2.

## Peer review information

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

## Acknowledgements

We thank our animal care technician, Annie Salesse, who cared for the mice used in this study, and Dr. Rafael Argüello for the SCENITH kit. LTM was supported by scholarships from the Fondation Armand-Frappier and the Fonds de Recherche du Québec – Santé (FRQS). We thank the Canadian Institutes of Health Research (PJT-190001 to SS.; PJT-148614 to KMH; PJT-175173 to JHF; PJT -168959 to DL) for financial support.

## Author contributions

**Linh Thuy Mai**: Conceptualization; Data curation; Formal analysis; Investigation; Writing—original draft. **Sharada Swaminathan**: Formal analysis; Investigation. **Trieu Hai Nguyen**: Formal analysis; Investigation. **Etienne Collette**: Data curation; Formal analysis; Investigation; Methodology. **Tania Charpentier**: Formal analysis; Investigation; Methodology. **Liseth Carmona-Pérez**: Investigation. **Hamza Loucif**: Methodology. **Alain Lamarre**: Resources; Methodology. **Krista M Heinonen**: Supervision. **David Langlais**: Resources; Supervision; Methodology. **Jörg H Fritz**: Data curation; Supervision; Methodology. **Simona Stäger**: Conceptualization; Resources; Data curation; Supervision; Funding acquisition; Methodology; Writing—original draft; Project administration.

Source data underlying figure panels in this paper may have individual authorship assigned. Where available, figure panel/source data authorship is listed in the following database record: biostudies:S-SCDT-10_1038-S44318-025-00485-2.

## Disclosure and competing interests statement

The authors declare no competing interests.

