## [Peer Review File · The EMBO Journal]

Transcription factor IRF-5 regulates lipid metabolism and mitochondrial function in murine CD8⁺ T-cells during viral infection

Linh Mai, Sharada Swaminathan, Trieu Nguyen, Etienne Collette, Tania Charpentier, Liseth Carmona-Perez, Hamza Loucif, Alain Lamarre, Krista Heinonen, David Langlais, Jörg Fritz, and Simona Stager

Corresponding author(s): *Simona Stager (simona.stager@inrs.ca)*

Review Timeline:

Submission Date:	6th Mar 24
Editorial Decision:	4th Apr 24
Appeal Received:	9th Dec 24
Editorial Decision:	5th Feb 25
Revision Received:	17th Mar 25
Editorial Decision:	27th Apr 25
Revision Received:	5th May 25
Accepted:	12th May 25

Editor: Ioannis Papaioannou

Transaction Report:

Dear Prof. Stager,

Thank you again for submitting your manuscript (EMBOJ-2024-117192) for consideration by The EMBO Journal. It has now been seen by three experts in the field, and we have received the full set of their comments, which are included below.

The referees recognize that this is an interesting and novel study of the IRF-5 function in CD8+ T cells during chronic viral infection. However, they also identify several technical limitations in the study and the manuscript, point out that the work remains largely descriptive and does not provide sufficient insights into the underlying regulation and mechanisms, and explain that some conclusions are not sufficiently supported by the presented data. In light of this input, I regret to say that we cannot further pursue publication of the manuscript in The EMBO Journal.

While we cannot pursue this manuscript further, we encourage you to transfer your study to our not-for-profit open-access sister journal, Life Science Alliance (LSA). We shared your manuscript and the accompanying reviews with LSA Executive Editor, Eric Sawey, who is interested in these findings, and would like to invite further consideration of this manuscript at LSA pending the following revisions:

- Address reviewer 1's specific comments except for #2, 14, 16, 17, 22, & 23. The minor points should be addressed.
- Address reviewer 2's comments.
- Address reviewer 3's comments.

We understand that such a revision might need to be re-reviewed, in which case, Dr. Sawey will walk the reviewers through the transfer process. We encourage you to use the link below to transfer your manuscript to LSA. You do not need to revise the manuscript before transferring it to LSA. Once you transfer, Dr. Sawey will email you an invitation to revise and resubmit, listing the same revision requests as mentioned above. Please feel free to reach out at e.sawey@life-science-alliance.org if you have any questions about the LSA journal, the transfer process or the revisions requested.

For The EMBO Journal, I am sorry we cannot be more positive on this occasion, but I nevertheless hope that you will find the referees' comments helpful in your work, and the option of transferring your manuscript to LSA worthwhile. I would like to thank you once again for your interest in our journal and for the opportunity to consider your manuscript.

Yours sincerely,

Referee #1:

The study by Mai et al investigates the role and function of IRF5 in LCMV-specific CD8 T cells during chronic viral infection. Using adoptive transfer of wt and IRF5-deficient LCMV -specific CD8 T cells, they report that IRF5 deficiency leads to loss of the cells during the course of the infection. In particular, KLRG1+ effector-like cells were lost in the absence of IRF5. Overall, the IRF5-deficient CD8 T cells expressed higher levels of coinhibitory receptors and displayed somewhat reduced effector functions compared to wt cells. Most strikingly, the IRF5-deficient cells showed an altered metabolism with reduced mitochondrial activity and increased ROS production upon in vitro activation. The authors speculate that the elevated ROS levels in IRF5-deficient CD8 T cells lead to lipid peroxidation and ensuing cell death.

This is an interesting study, since very little knowledge exists about the role of IRF5 in CD8 T cells. IRF-5 is a transcription factor that was reported to play a critical role in a multitude of different processes including tumour suppression, cell cycle regulation, M1 macrophage polarization and many more. Furthermore, in previous studies IRF-5 was shown to initiate the assembly of the TCR in CD4 T cells during chronic infection and to regulate CD4 T cell migration to the lymph node as well as the induction of Th1- and Th17-associated cytokines. In addition, the authors have previously reported that IRF-5 acts downstream of TLR-7 in CD4 T cells and induces cell death in the context of chronic infections.

Despite the clear phenotypes observed when comparing wt and IRF5-deficient CD8 T cells in the context of chronic LCMV infection, the study largely remains on a descriptive level and fails to delineate the mechanisms that lead to the reduced maintenance of the IRF5-deficient CD8 T cells and to the selective abrogation of the KLRG1+ population of CD8 T cells. While the data appears convincing and is presented in a logical structure, important control experiments as well as experiments directly linking the changes induced by IRF-5 deficiency to a regulatory mechanism are missing. As detailed below, the manuscript would benefit from some revisions to complement missing information and to provide for a more holistic study of the role of IRF-5 in CD8 T cells. Finally, performing additional experiments could aid the mechanistic insights.

Specific comments

1. The authors observe strongly reduced ATP induction in IRF5-deficient CD8 T cells upon in vitro activation (Figure 6F). At the same time. The authors observe faster proliferation of IRF5-deficient CD8 T cells (Figure 4). How can these two seemingly opposing observations be explained?
2. The authors focus their analysis of the role of IRF5 in CD8 T cells during chronic LCMV infection. Is IRF 5 also expressed in CD8 T cells after acute infection and does it play a role in the differentiation and maintenance of CD8 T cells in acute infection?
3. Throughout the study the authors consider 30d p.i. as the "chronic stage" of the infection and continue to discriminate effector and memory cells based on the expression of KLRG1 and CD127. However, previous studies showed that the differentiation of CD8 T cells during a chronic infection is different compared to an acute infection. Therefore, the discrimination based on these markers might not be appropriate. In addition, throughout the study the reference to T cell exhaustion is missing even though the authors study a chronic infection.

It is somewhat surprising that the authors observe such a large percentage of KLRG1+ CD8 T cells at later stages of chronic infection. It has been reported that these KLRG1+ cells develop early during chronic infection and resemble an early but transient differentiation of effector-like cells. In Figure 2D it is also shown that there is a reduction of KLRG1 expression in wt cells between d8 and day 30. However, this is followed by a strong upregulation of KLRG1 expression at day 60. Might this be due to the fact that the mice have controlled the infection by day 60?
4. The authors propose that IRF-5 prevents functional exhaustion based on their in vitro restimulation experiments and by looking at the expression levels of co-inhibitory receptors. On the other hand, the authors might consider that IRF-5 promotes the cytotoxic potential of p14 cells, which is indeed the case when looking at the cytokine expression after restimulation of cells 8d p.i. As a result, the cells might upregulate the expression of co-inhibitory receptors and undergo cell death as a protective mechanism to combat immunopathology. In addition, it seems that the IRF-5 deficient p14 cells rather differentiate into CD127+ cells compared to the WT cells, raising the hypothesis whether IRF-5 affects the differentiation of the p14 cells.
5. Figure 2D: comparison of the percentage of KLRG1+ wt cells at day 8 and day 30: the presented FACS data show a decline in frequencies, but this decline is not evident in the summary graph.
6. Figure S1: why are there such large differences between male and female mice in terms of survival? Why are so many male mice receiving ko P14 cells succumbing to the infection?
7. Figure 2: Can the authors exclude that the ko P14 cells are rejected at later time points?
8. Figure S2: endogenous response of wt and IRF5-ko mice: it would be important to gate on LCMV-specific CD8 T cells using tetramers.
9. Overall, it would be important to show viral titers in the experiments (e.g. Fig .2, Fig.3) to evaluate whether viral loads are comparable in recipients of wt or IRF5-deficient P14 cells and at which time point the chronic infection is resolved in the applied experimental setting.
10. Figure 3: The percentage of IFN γ -producing cells at day 30 seems rather large (75% of wt cells). Are the mice still chronically infected at this time point?
11. Page 9: What is meant by "IRF-5 was shown to compete with IRF-4"? Competition for DNA binding sites?
12. Figure 3: at day 8 the percentage of KLRG1+ cells is increased in the ko compared to wt cells, but this does not seem to be the case for CX3CR1+ cells. How is this difference explained?
13. Figure S5: Wt mice seem to control the infection between day 18 and 45, as also reflected by the decline in PD-1 expression on LCMV-specific CD8 T cells. This might also play an important role in the difference phenotypes and function of the CD8 T cells in wt and ko hosts.
14. Figure 4: Why was bulk sequencing done instead of scRNA sequencing? Bulk sequencing "only" reveals the average transcriptional profile of the cells and offers no possibility to resolve the transcriptomic analysis on a subset level.
15. Figure 4: What about EdU staining to see whether cells are still actively proliferating at the respective time points post infection?
16. Figure 4: How does this look during an acute infection? Since you see the same phenotype upon stimulation in vitro: Is the phenomenon you observe something specific during T cell exhaustion or just in general the function of IRF-5?
17. It would be interesting to analyze the differentiation of Wt and IRF5-deficient T_{pex} in vivo upon adoptive transfer into

infection matched hosts with respect to their relative differentiation trajectories and numeric maintenance.

18. Figure 5: Data provided does not make the point that IRF-5 deficient CD8 T cells undergo cell death. The authors should show cell death on protein level by staining for example for apoptosis markers or other markers to study the cell death mechanism that potentially leads to the observed decline in cells. In turn, the authors could design an experiment, in which they block the respective cell death pathway and thereby achieve the retention of the cells.

19. Figure 6: Are these experiments performed with naïve cells? How would the analysis look for cells isolated from a chronic infection? Are the cells metabolically different at 8 or 30 days post infection? What is the link of the seahorse analysis to Figure 5 where most emphasis is given on the lipid metabolism?

20. In the metabolic analyses, why is there a difference between male and female cells?

21. Figure 7: If lipid peroxidation leads to death of the cells, why does it primarily affect the KLRG1+ effector-like cells? As T_{pex} also rely on fatty acid metabolism, why is this subset not compromised in absence of IRF5?

22. The authors propose that the mitochondrial envelope of IRF-5 deficient cells has severe defects, and that the cells possibly die by lipid peroxidation. As the authors make this assumption based on only the staining with TMRM and the ratio of 590/510 nm, additional data to prove their hypothesis is missing. For example, the authors might look at TEM images to have a look at the ultrastructure of the mitochondria, perform light/confocal microscopy to count the number of mitochondria in the cells. In addition, the authors might perform metabolomics to see whether metabolites involved in lipid peroxidation are enriched. To make the link between cell death and lipid peroxidation the authors might perform an experiment, in which they disrupt the pathway of lipid peroxidation to see whether this would be sufficient to rescue the cells. To this end, the authors might want to inhibit lipid peroxidation to directly show an involvement in cell death of the IRF5-deficient CD8 T cells by applying anti-oxidant treatment such as with vitamin E or the recently described arylthiazine derivative (<https://doi.org/10.1038/s41598-021-81741-3>). While the data looks very convincing, a deeper mechanistic understanding could aid in understanding of how IRF-5 is involved in metabolic rewiring.

23. In general, it would be helpful to use the "commonly" used markers to identify T_{pex} (e.g. Tcf1), effector-like cells (e.g. CX3CR1) and terminally exhausted cells (CXCR6, CD101) to assess their relative abundance percentage-wise but also numerically and to assess the function and metabolism based on the subtype of cells.

Minor points

- Figure 1: Please specify in the manuscript that you are performing a chronic infection.
- Figure 1A: When showing the frequencies of IRF-5+ CD8 T cells, are the plots pre-gated on p14 cells / CD45.2+ cells? Please clarify.
- Figure 1A: Please show representative flow cytometry plot for gating on CD45.2+ p14 cells.
- Figure 1B: Please show representative flow cytometry plot for gating on KLRG1+CD127- p14 cells.
- Figure 1C: Please show representative flow cytometry plot for gating on CD44+CD127+ p14 cells.
- Figure 1, Supplement Figure 2: Naïve T cells are not KLRG1+ nor CD127+. Please either change labelling of "naïve" into d0 or look at IRF-5 expression in CD44- KLRG1- and CD127- p14 cells.
- Figure 2B and 2C: Labelling of the y-axis should be "% of p14 CD8 T cells..." or "# of p14 CD8 T cells...". Please correct.
- Supplemental Figure 2: The CD8 T cells investigated here are still transgenic. The only difference is that they are not specific for the virus-specific glycoprotein gp33. Please correct.
- Figure 7D-H: Why was analysis done on day 15 p.i.? Please specify or perform analysis on day 21 p.i. similar to RNA-Seq data.

Referee #2:

The authors aim to elucidate the role of transcription factor IRF-5 in the metabolic rewiring of CD8 T cells during chronic infections.

The manuscript provides evidence that IRF-5 is expressed in murine CD8 T cells and that that lack of IRF-5 in CD8 T cells results in a profoundly reduced number of CD8 T cells during chronic infection in mice.

However, the subsequent parts of the manuscript represent an incoherent series of data, time points, and experimental models and that are not sufficient to draw scientific conclusions. This includes the following points:

- On page 7 the authors state that there were substantial differences between male and female mice, which made them investigate sex-specific results. These are however not reported throughout the whole manuscript.
- More specific, Fig 2/3 is male only, 4/5 is male/female, Fig 6 male only, Fig 7 male/female - why these differences?
- % male/female in Figures 4/5/7 in unspecified
- Is there an explanation for the reported differences between male/female (Figure 1)?
- Time points, experimental models and sex of mice are not comparable throughout the results. This may lead to the suspicion that the missing data/ time points do not fit into the data presented.

- Fig 3 Cytokine expression does not represent functional exhaustion
- Fig 3 reports d30 pi vs. Fig 4 reports d21 pi; please state the reason?
- Fig 5 does not backup any functional statement about cell death or metabolic defects, it only represents transcriptomics that may lead to the generation of a hypothesis
- Have you measured apoptosis/necrosis?
- Fig 6 cells reports data without adoptive transfer, cells are stimulated over 6/24h - how does this represent the metabolic fingerprint of exhausted T cells? How does this correlate with the other time points/ experimental model of murine chronic infection.
- Fig 6: how is statistics performed when only one experiment out of three is shown? N=4 technical replicates?
- Fig 7 mitochondrial envelope: that is something that can be hypothesized on due to RNA data, data is yet insufficient to draw any conclusions; protein expression of OXPHOS/ mitochondrial mass and other investigations may be required
- Fig 7 reports different time points itself: d21 (B/C) and d15 pi (D-H), what is the reason or the scientific basis for this?
- Seahorse data shows almost complete loss of metabolic capacity; T cells are unlikely to survive this; this may not represent the functional metabolic situation in the model of chronic infection
- Reduced TMRM fluorescence does not show "cells that are metabolically stressed"; do you have used uncoupling agents as a negative control?
- 5 μ M MitoSOX leads to DNA staining and does not represent mROS, <1 μ M is required!
- How should mROS (if correctly measured) contribute to this matter? Of note, mROS are required for T cell activation
- Data on lipid long chain fatty acid uptake is incongruent between Fig 6/7, further highlighting the different experimental models
- The authors do not specify "metabolic rewiring" during chronic infection, which makes it difficult to identify "changes" to metabolic rewiring
- How is lipid peroxidation involved in this matter?
- Please specify the concept of "mitochondrial envelope"
- Please clearly state in the abstract and in the introduction that only murine cells are being investigated
- Several pages of the results describe the names and context of transcripts that are irrelevant for the study, please check and remove to keep the overall view
- Since the role of IRF5 in cell cycle arrest has already been described, it remains elusive why the authors examine this subject again and provide a complete figure about it
- Given the tremendous amount of transcription factors, it should be clarified what the value of IRF5 is in the overall concept of CD8 T cell exhaustion
- The introduction is rather a review article itself and should be significantly shortened

Referee #3:

Mai and colleagues analyze the role played by the nuclear factor IRF5 in the phenomenon of T cell exhaustion following infection with clone 13 LCMV. They present data supporting a role played by IRF5 in the limitation of T cell functional exhaustion. Though TcR-transgenic virus-specific IRF5-deficient CD8 T cells exhibited a better proliferative capacity in the acute phase of infection, they were prone to apoptosis, exhibited deeper exhaustion in the chronic phase and were unable to control virus replication. They also provide evidence, mostly based on gene expression analysis and by in vitro metabolic inhibition, that the absence of IRF5 expression limits glycolysis, promote oxidative stress accompanied by lipid peroxidation.

One of the weakness of this work is that it is very descriptive and does not present an in-depth description of the mechanisms underlying IRF5 expression.

I have a few remarks that need authors' attention:

- Legend of Figure 1 must be modified. In the legend it is said that "Graph shows ... the frequency (right) of p14 CD8 T cells present in the spleen of infected mice." To be correct, it should be : "Graph shows ... the frequency (right) of p14 CD8 T cells expressing IRF5 present in the spleen of infected mice."
- page 6, first paragraph of the results section, it is said that "The kinetics of IRF-5 expression in CD127+ CD8+ T cells was slightly different. Indeed, higher frequencies of IRF-5+ CD127+ cells were observed between d8 and 34p.i. (Figure 1C)." This is not the case since at day 15 the percentage of IRF5-expressing cells among CD127+ cells is half that at day 8. Please modify.
- Same page, same paragraph, it is also said "IRF-5 expression was found to co-localize with the nucleus in CD8 T cells during the whole course of infection, with 60-70% of the cells expressing IRF-5 in the nucleus between d8 and 60 p.i. (Figure 1D)." When I look at the graph, I only see 50 to 70%. Please modify.
- Page 7: "To determine the function of IRF-5 in CD8 T cells, we generated IRF-5-deficient p14 transgenic mice (Irf5fl/flx CMV-Cre+ p14), which have CD8 T cells that are specific for the LCMV gp33 antigen". To be fair, it should be specified, here or in the Mat and Methods, that, because of Cre being expressed under CMV promoter, ablation of IRF5 expression in these mice is ubiquitous and not restricted to T cells. The origin of p14 mice should also be mentioned.
- The authors observed a difference of survival after infection between mice transferred with p14 T cells from male or female donors (Fig S1). Since, in C57BL/6 mice, females develop immunity against male antigens, it should be specified whether p14 T cells were transferred in sex-matched recipients.

- I am a bit concerned about the experiments analyzing the metabolism of IRF^{-/-} p14 CD8 T cells. These experiments were carried out after stimulating in vitro naive T cells with specific antigenic peptide. Results show that IRF^{-/-} CD8 T cells displayed deeply impaired glycolytic and oxidative metabolisms 24 hr after stimulation (see Figure 6). This is surprising to me since increased metabolism is important for T cell proliferation and, according to Figure 2C, the same cells appear to have no problem expanding in the acute phase of LCMV infection (3000 p14 T cells were injected per mouse and more than 7 millions of them were recovered 8 days after infection). Can the authors comment on this. Have the authors analyzed naive IRF^{-/-} p14 CD8 T cells for in vitro proliferation after peptide stimulation?

- Fig 5A and 5B display expression of Cell Death genes and Exhaustion genes, respectively. It is the opposite in the legend of the figure. Please correct.

Since analysis of mitochondrial ROS production did not show any significant differences between IRF5^{-/-} and WT T cells (Fig 7E), the data should not be presented as supporting the presence of mitochondrial stress in IRF5^{-/-} T cells (page 16).

- Please check the references. In the discussion section page 18, line 4, reference #58 is given for the role of TOX in T cell development while the study presented there talks about the assembly of mitochondrial complex 1.

** As a service to authors, EMBO Press provides authors with the possibility to transfer a manuscript that one journal cannot offer to publish to another EMBO publication or the open access journal Life Science Alliance launched in partnership between EMBO Press, Rockefeller University Press and Cold Spring Harbor Laboratory Press. The full manuscript and if applicable, reviewers' reports, are automatically sent to the receiving journal to allow for fast handling and a prompt decision on your manuscript. For more details of this service, and to transfer your manuscript please click on Link Not Available. **

We thank all three reviewers for their valuable comments that have significantly strengthened the manuscript. Please find below a point-by-point response to all raised concerns.

Reviewer #1

“The study by Mai et al investigates the role and function of IRF5 in LCMV-specific CD8 T cells during chronic viral infection. Using adoptive transfer of wt and IRF5-deficient LCMV -specific CD8 T cells, they report that IRF5 deficiency leads to loss of the cells during the course of the infection. In particular, KLRG1+ effector-like cells were lost in the absence of IRF5. Overall, the IRF5-deficient CD8 T cells expressed higher levels of coinhibitory receptors and displayed somewhat reduced effector functions compared to wt cells. Most strikingly, the IRF5-deficient cells showed an altered metabolism with reduced mitochondrial activity and increased ROS production upon in vitro activation. The authors speculate that the elevated ROS levels in IRF5-deficient CD8 T cells lead to lipid peroxidation and ensuing cell death.

This is an interesting study, since very little knowledge exists about the role of IRF5 in CD8 T cells. IRF-5 is a transcription factor that was reported to play a critical role in a multitude of different processes including tumour suppression, cell cycle regulation, M1 macrophage polarization and many more. Furthermore, in previous studies IRF-5 was shown to initiate the assembly of the TCR in CD4 T cells during chronic infection and to regulate CD4 T cell migration to the lymph node as well as the induction of Th1- and Th17-associated cytokines. In addition, the authors have previously reported that IRF-5 acts downstream of TLR-7 in CD4 T cells and induces cell death in the context of chronic infections.

Despite the clear phenotypes observed when comparing wt and IRF5-deficient CD8 T cells in the context of chronic LCMV infection, the study largely remains on a descriptive level and fails to delineate the mechanisms that lead to the reduced maintenance of the IRF5-deficient CD8 T cells and to the selective abrogation of the KLRG1+ population of CD8 T cells. While the data appears convincing and is presented in a logical structure, important control experiments as well as experiments directly linking the changes induced by IRF-5 deficiency to a regulatory mechanism are missing. As detailed below, the manuscript would benefit from some revisions to complement missing information and to provide for a more holistic study of the role of IRF-5 in CD8 T cells. Finally, performing additional experiments could aid the mechanistic insights.

1. The authors observe strongly reduced ATP induction in IRF5-deficient CD8 T cells upon in vitro activation (Figure 6F). At the same time. The authors observe faster proliferation of IRF5-deficient CD8 T cells (Figure 4). How can these two seemingly opposing observations be explained?

In our in vitro model, p14 cells only receive TCR stimulation, so the phenomenon we observe is merely the IRF-5 effect on the cell metabolism through the TCR signaling pathway. Cells in vivo receive several additional signals, such as costimulation and cytokine signalling, that could compensate for the lack of IRF-5 function by inducing other transcription factors. One of those is IRF-4, which competes for DNA binding with IRF-5 and is known to promote glycolysis. We agree that this was the major weakness in our first submission. For this reason, we have now analyzed the cellular metabolic activity in adoptively transferred WT and KO p14 cells ex-vivo (New Figures 7A-G).

2. The authors focus their analysis of the role of IRF5 in CD8 T cells during chronic LCMV infection. Is IRF 5 also expressed in CD8 T cells after acute infection and does it play a role in the differentiation and maintenance of CD8 T cells in acute infection?

This is a great point. We initially investigated IRF-5 function in CD8 T cells in mice infected with recombinant vaccinia virus expressing the ovalbumin peptide SIINFEKL. To this end, we adoptively transferred OT-I cells into congenic mice and then infect them with VV-SIINFEKL. We first assessed IRF-5 expression in WT OT-I cells (s. below, A) and then followed WT and IRF-5-deficient cells for 60 days. As shown below (B-F), we did not observe any difference between WT and KO cells, suggesting the IRF-5 function in an acute infection setting is redundant and only plays a major role when cells are exhausted.

We have also carried out similar adoptive transfer experiments as described in the manuscript, but this time, we infected mice with LCMV Armstrong. This experiment is still ongoing, but we are showing results for d5 and 8 p.i. below. We did not observe any difference in the percentage and number of adoptively transferred cells in both groups (A-B), and only saw small differences in the percentage of KLRG1⁺ cells, as well as a reduction in the frequency of memory-like CD127⁻ KLRG1⁺ cells, but only in male mice (C, D, E). Moreover, there was no difference in the frequency or MFI of (F) IFN γ - and (G) TNF-expressing cells. We also looked at the cells' capacity to internalize lipids (H) and glucose (I) and measured mitochondrial ROS (J) and TMRM (K). Finally, we checked the cell cycle (L).

3. Throughout the study the authors consider 30d p.i. as the "chronic stage" of the infection and continue to discriminate effector and memory cells based on the expression of KLRG1 and CD127. However, previous studies showed that the differentiation of CD8 T cells during a chronic infection is different compared to an acute infection. Therefore, the discrimination based on these markers might not be appropriate. In addition, throughout the study the reference to T cell exhaustion is missing even though the authors study a chronic infection.

We agree with Reviewer #1 that using KLRG1 and CD127 to differentiate subpopulations of CD8 T cells during chronic infection does not accurately depict the various CD8 subpopulations present during chronic stages of infection. For this reason, we used the term effector-like cells for KLRG1⁺CD127⁻ or memory-like cells for KLRG1⁻CD127⁺ throughout the text. Moreover, in the initial version of the manuscript, we also stained for other markers (CX3CR1 and CD101) to discriminate between the various exhausted subpopulations (Figure 3 J, K, L). We have now added another staining using Ly108 and CD69 to characterize CD8 T cells (Supplemental Figures 3F-J).

It is somewhat surprising that the authors observe such a large percentage of KLRG1+ CD8 T cells at later stages of chronic infection. It has been reported that these KLRG1+ cells develop early during chronic infection and resemble an early but transient differentiation of effector-like cells. In Figure 2D it is also shown that there is a reduction of KLRG1 expression in wt cells between d8 and day 30. However, this is followed by a strong upregulation of KLRG1 expression at day 60. Might this be due to the fact that the mice have controlled the infection by day 60?

Yes, it is possible that this upregulation reflects the decrease in antigen levels present in the spleen at d60p.i.. At this later stage of infection, the virus is cleared in most organs/tissues (including the spleen); however, it persists in the kidneys and brain.

4. The authors propose that IRF-5 prevents functional exhaustion based on their in vitro restimulation experiments and by looking at the expression levels of co-inhibitory receptors. On the other hand, the authors might consider that IRF-5 promotes the cytotoxic potential of p14 cells, which is indeed the case when looking at the cytokine expression after restimulation of cells 8d p.i. As a result, the cells might upregulate the expression of co-inhibitory receptors and undergo cell death as a protective mechanism to combat immunopathology. In addition, it seems that the IRF-5 deficient p14 cells rather differentiate into CD127+ cells compared to the WT cells, raising the hypothesis whether IRF-5 affects the differentiation of the p14 cells.

We did not observe any difference in the frequency of Granzyme B+ WT and KO p14 cells between d8 and 60 p.i. (s. below Figure 2A and B).

A Male mice

B Female mice

We do not think that IRF-5-deficiency affects CD8 T cell differentiation. There are more CD127+ memory-like cells because they are less prone to cell death. If KO cells were preferentially differentiating into memory-like cells, we would not see the dramatic decrease in cell numbers that occurs after d8 p.i..

5. Figure 2D: comparison of the percentage of KLRG1+ wt cells at day 8 and day 30: the presented FACS data show a decline in frequencies, but this decline is not evident in the summary graph. We agreed with the Reviewer #1. We choose a better representative FACS plot for this panel.

6. Figure S1: why are there such large differences between male and female mice in terms of survival? Why are so many male mice receiving ko P14 cells succumbing to the infection?

We actually do not know. We think that male mice are more pro-inflammatory than females and perhaps were dying because of excessive inflammation. However, we did not look further for possible mechanisms. This is an interesting question that may also be relevant for other infections, such as COVID19.

7. Figure 2: Can the authors exclude that the ko P14 cells are rejected at later time points?

The KO cells are genetically identical to the WT cells. The only difference is that they do not express the Cre recombinase. As mentioned in the text, WT cells are not from p14 mice, but from *Irf5*^{flox/flox} x *CMV-Cre*⁻ p14 mice.

8. Figure S2: endogenous response of wt and IRF5-ko mice: it would be important to gate on LCMV-specific CD8 T cells using tetramers.

We agree with Reviewer #1. We added this information in the Supplemental Figure 2 (S Figures 2D-F).

9. Overall, it would be important to show viral titers in the experiments (e.g. Fig .2, Fig.3) to evaluate whether viral loads are comparable in recipients of wt or IRF5-deficient P14 cells and at which time point the chronic infection is resolved in the applied experimental setting.

We usually transfer 2000 p14 cells per mouse. This small number of cells has no impact on the viral burden in mice. We have added the viral titers in various organs below (s. Figure 3 A-E).

10. Figure 3: The percentage of IFN γ -producing cells at day 30 seems rather large (75% of wt cells). Are the mice still chronically infected at this time point?

Yes, the virus is still present in most tissues/organs.

11. Page 9: What is meant by "IRF-5 was shown to compete with IRF-4"? Competition for DNA binding sites?

Yes. We have now modified the text accordingly.

12. Figure 3: at day 8 the percentage of KLRG1+ cells is increased in the ko compared to wt cells, but this does not seem to be the case for CX3CR1+ cells. How is this difference explained?

We do not know for sure, since no study in the literature compared the two set of staining. It is possible that KLRG1 cells include effector-like and terminally exhausted cells, which are also a sort of effector cells. This could explain this discrepancy.

13. Figure S5: Wt mice seem to control the infection between day 18 and 45, as also reflected by the decline in PD-1 expression on LCMV-specific CD8 T cells. This might also play an important role in the difference phenotypes and function of the CD8 T cells in wt and ko hosts.

We do not think that this is an issue. T cell-specific IRF-5 deficient mice are incapable to control infection because their CD8 T cells are less functional. The viral load in mice adoptively transferred with WT or KO p14 cells supports our interpretation. Indeed, IRF-5 deficient p14 cells are defective and die even when viral loads are comparable (s. Figure 3 above).

14. Figure 4: Why was bulk sequencing done instead of scRNA sequencing? Bulk sequencing "only" reveals the average transcriptional profile of the cells and offers no possibility to resolve the transcriptomic analysis on a subset level.

We agree with Reviewer #1. scRNA seq would have been much better. The only reason why we did not perform scRNA seq is merely financial. These experiments were not budgeted in the initial grant.

15. Figure 4: What about EdU staining to see whether cells are still actively proliferating at the respective time points post infection?

We have not done this type of analysis, because we think that it will not add much to the story. In Figure 4 we show that IRF-5-deficient p14 cells enter the cell cycle in greater proportion and in Figure 5 we now show that they also die in greater proportion.

16. Figure 4: How does this look during an acute infection? Since you see the same phenotype upon stimulation in vitro: Is the phenomenon you observe something specific during T cell exhaustion or just in general the function of IRF-5?

This is a very important point that we have addressed above (#2).

17. It would be interesting to analyze the differentiation of Wt and IRF5-deficient T_{pex} in vivo upon adoptive transfer into infection matched hosts with respect to their relative differentiation trajectories and numeric maintenance.

Although this type of experiment would of great interest, it is practically nearly impossible to do, because of the unethically large number of mice we would need. In mice that received WT cells, there are 2-3 million p14 cells present in the spleen at d30p.i., 50% of which are progenitor-like T_{pex}. In the case of KO cells, we only see less than 1 million p14 cells per spleen at d30p.i. (s. Figure 2).

18. *Figure 5: Data provided does not make the point that IRF-5 deficient CD8 T cells undergo cell death. The authors should show cell death on protein level by staining for example for apoptosis markers or other markers to study the cell death mechanism that potentially leads to the observed decline in cells. In turn, the authors could design an experiment, in which they block the respective cell death pathway and thereby achieve the retention of the cells.*

We agree with Reviewer #1. We have now stained our cells for Annexin V and cleaved caspase 3 and added these results to the manuscript (s. Figure 5A and B).

19. *Figure 6: Are these experiments performed with naïve cells? How would the analysis look for cells isolated from a chronic infection? Are the cells metabolically different at 8 or 30 days post infection? What is the link of the seahorse analysis to Figure 5 where most emphasis is given on the lipid metabolism?*

Reviewer #1 makes a good point. Because of the extremely low number of cells present in the spleen at d21 and 30 p.i. (we had to pool 7-17 mice for 1 RNAseq sample), we were not able to perform Seahorse analysis on ex-vivo purified cells. For this reason, we turned to a recently described methods called SCENITH (Arguello et al, Cell Metabolism 2020) that measures cell metabolism and can be performed by flow cytometry. The disadvantage of this method is that it is indirect; however, it allows to differentiate between various CD8 T cell subpopulations. These new data are now part of Figure 7 (A-G).

20. *In the metabolic analyses, why is there a difference between male and female cells?*

Unfortunately, we do not have an answer to this question. We repeated this experiment several times and always observed the same difference between male and female cells.

21. *Figure 7: If lipid peroxidation leads to death of the cells, why does it primarily affect the KLRG1+ effector-like cells? As T_{pex} also rely on fatty acid metabolism, why is this subset not compromised in absence of IRF5?*

We do not know. It is possible that it is a question of chromatin accessibility that dictates the various IRF-5 functions, or the type of concomitant signal received by the cells. This is an important question that we would like to explore in the near future.

22. *The authors propose that the mitochondrial envelope of IRF-5 deficient cells has severe defects, and that the cells possibly die by lipid peroxidation. As the authors make this assumption based on only the staining with TMRM and the ratio of 590/510 nm, additional data to prove their hypothesis is missing. For example, the authors might look at TEM images to have a look at the ultrastructure of the mitochondria, perform light/confocal microscopy to count the number of mitochondria in the cells. In addition, the authors might perform metabolomics to see whether metabolites involved in lipid peroxidation are enriched. To make the link between cell death and lipid peroxidation the authors might perform an experiment, in which they disrupt the pathway of lipid peroxidation to see whether this would be sufficient to rescue the cells. To this end, the authors might want to inhibit lipid peroxidation to directly show an involvement in cell death of the IRF5-deficient CD8 T cells by applying anti-oxidant treatment such as with vitamin E or the recently described arylthiazine derivative (<https://doi.org/10.1038/s41598-021-81741-3>). While the data looks very*

convincing, a deeper mechanistic understanding could aid in understanding of how IRF-5 is involved in metabolic rewiring.

We agree with Reviewer #1 that we should solidify our hypothesis by providing further mechanistic insights and thank Reviewer #1 for their mindful suggestions. To this end, we performed TEM on in vitro stimulated WT and KO cells and analyzed various parameters, such as number of mitochondria, mitochondria perimeter, number of cristae per mitochondria, cristae volume density and MERC length. We found that IRF-5 deficiency resulted in fewer mitochondria per cell, but mitochondria were larger. We also found a reduced volume and number of cristae per mitochondrion. These results support our hypothesis that the absence of IRF-5 alters mitochondrial functions and are now part of Figure 6. Regarding the other suggested experiments, we are planning, in the near future, to further dissect in detail the pathways by which IRF-5 contributes to regulate the lipid metabolism of CD8 T cells.

23. In general, it would be helpful to use the "commonly" used markers to identify T_{pex} (e.g. Tcf1), effector-like cells (e.g. CX3CR1) and terminally exhausted cells (CXCR6, CD101) to assess their relative abundance percentage-wise but also numerically and to assess the function and metabolism based on the subtype of cells.

We have now additionally stained cells with Ly108 and CD69 to distinguish between Tex progenitor 1 and progenitor 2, and intermediate and terminally exhausted cells. These results are shown in Supplemental Figures 3F-J and confirm our previous results. We found significantly lower frequencies of intermediate exhausted cells and higher frequencies of terminally exhausted cells. No significant differences between WT and KO were observed for Tex progenitor 1 and 2 with exception of d60p.i, when KO cells displayed higher frequencies of Tex progenitor 1.

We also used the same markers to assess the metabolic activity of ex-vivo adoptively transferred cells (Figure 7A-G). Taken together, the new data demonstrate that IRF-5 deficiency mostly affects effector-like and terminally exhausted cells.

You can find below the cell numbers for the various CD8 T cell populations. We can add them to the manuscript if required.

Minor points

• Figure 1: Please specify in the manuscript that you are performing a chronic infection. The text was modified accordingly.

• *Figure 1A: When showing the frequencies of IRF-5+ CD8 T cells, are the plots pre-gated on p14 cells / CD45.2+ cells? Please clarify.*

We have clarified this point in the figure legend.

• *Figure 1A: Please show representative flow cytometry plot for gating on CD45.2+ p14 cells. Please find the requested plots below. We can add them to the manuscript if required.*

• *Figure 1B: Please show representative flow cytometry plot for gating on KLRG1+CD127- p14 cells. Please find the requested plots below. We can add them to the manuscript if required.*

Please find the requested plots below. We can add them to the manuscript if required.

• *Figure 1C: Please show representative flow cytometry plot for gating on CD44+CD127+ p14 cells. Please find the requested plots below. We can add them to the manuscript if required.*

• *Figure 1, Supplement Figure 2: Naïve T cells are not KLRG1+ nor CD127+. Please either change labelling of "naïve" into d0 or look at IRF-5 expression in CD44- KLRG1- and CD127- p14 cells.*

We have now changed all the graphs to d0, as suggested.

• *Figure 2B and 2C: Labelling of the y-axis should be "% of p14 CD8 T cells..." or "# of p14 CD8 T cells...". Please correct.*

Thank you for the suggestion.

• *Supplemental Figure 2: The CD8 T cells investigated here are still transgenic. The only difference is that they are not specific for the virus-specific glycoprotein gp33. Please correct.*
We modified the text accordingly.

• *Figure 7D-H: Why was analysis done on day 15 p.i.? Please specify or perform analysis on day 21 p.i. similar to RNA-Seq data.*

We have now repeated all results and analysed all parameters at d15 and 21p.i.

Reviewer#2

The authors aim to elucidate the role of transcription factor IRF-5 in the metabolic rewiring of CD8 T cells during chronic infections. The manuscript provides evidence that IRF-5 is expressed in murine CD8 T cells and that that lack of IRF-5 in CD8 T cells results in a profoundly reduced number of CD8 T cells during chronic infection in mice. However, the subsequent parts of the manuscript represent an incoherent series of data, time points, and experimental models and that are not sufficient to draw scientific conclusions. This includes the following points:

1) On page 7 the authors state that there were substantial differences between male and female mice, which made them investigate sex-specific results. These are however not reported throughout the whole manuscript. More specific, Fig 2/3 is male only, 4/5 is male/female, Fig 6 male only, Fig 7 male/female - why these differences?

% male/female in Figures 4/5/7 in unspecified

Is there an explanation for the reported differences between male/female (Figure 1)?

Time points, experimental models and sex of mice are not comparable throughout the results. This may lead to the suspicion that the missing data/ time points do not fit into the data presented.

We actually provided data on male and female mice throughout the manuscript. Data in the main figure for one sex were always complemented by data for the other sex in the supplemental figures. To avoid results redundancy, we often show the most important graphs (we already have 7 main figures and 7 supplemental figures). The full analysis is shown when differences are observed between both sexes.

2) Fig 3 Cytokine expression does not represent functional exhaustion

We agree with Reviewer #2 that cytokine expression alone does not represent functional exhaustion, but reduction in cytokine expression coupled with increased expression of inhibitory receptors does.

3) Fig 3 reports d30 pi vs. Fig 4 reports d21 pi; please state the reason?

We decided to chose d21, because at d30 the number of *Irf5*^{-/-} p14 cells would be far too low for performing RNAseq of ex-vivo purified cells. Already at d21, we had to pool 7-17 mice per sample to be able to have enough cells.

4) Fig 5 does not backup any functional statement about cell death or metabolic defects, it only represents transcriptomics that may lead to the generation of a hypothesis

We agree with Reviewer #2. For this reason, we have now added a staining for Annexin V and cleaved caspase 3 as Figures 5A and B and modified Figure 5,6, and 7.

5) *Have you measured apoptosis/necrosis?*

We did now (s. Figure 5A and B), thank you for the suggestion.

6) - *Fig 6 cells reports data without adoptive transfer, cells are stimulated over 6/24h - how does this represent the metabolic fingerprint of exhausted T cells? How does this correlate with the other time points/ experimental model of murine chronic infection.*
Reviewer #2 is right. Because of the extremely low number of adoptively transferred WT and KO p14 cells present in the spleen during the chronic stages of infection, it is impossible to perform Seahorse analysis on ex-vivo purified cells. We have now used the SCENITH kit, which indirectly assess the metabolic activity of cells (Arguello et al., Cell Metabolism 2020). These results are now part of Figure 7 (A-G).

7) *Fig 6: how is statistics performed when only one experiment out of three is shown? N=4 technical replicates?*

We used 4 different mice as donors and treated the cells from each mouse independently. The 4 mice represent n=4. This means that we had technical replicates for each mouse and then 4 different cell donors. This was repeated 3 times.

8) *Fig 7 mitochondrial envelope: that is something that can be hypothesized on due to RNA data, data is yet insufficient to draw any conclusions; protein expression of OXPHOS/ mitochondrial mass and other investigations may be required.*

We agree with Reviewer #2, and we are grateful for their suggestion. We have now added new data to support our hypothesis: we performed TEM of in vitro activated WT and KO p14 cells and analyzed various parameters, including the number and size of mitochondria (Figure 6 I-O); we also assessed the mitochondrial mass in vivo using Mito Tracker Green.

9) *Fig 7 reports different time points itself: d21 (B/C) and d15 pi (D-H), what is the reason or the scientific basis for this?*

We agree with Reviewer #2 and apologize for the mistake. We have now replaced the graphs with new data analyzed at d15 and 21p.i..

10) *Seahorse data shows almost complete loss of metabolic capacity; T cells are unlikely to survive this; this may not represent the functional metabolic situation in the model of chronic infection*

We agree with Reviewer #2. When we stimulate p14 cells in vitro, we only stimulate them through the TCR. However, in vivo, the cells receive other stimuli, e.g. through costimulation or cytokine receptors. Furthermore, the cells are in a chronic inflammatory environment, which is quite different than the optimized tissue culture conditions. For this reason, we have now added new data on the ex-vivo metabolic activity of adoptively transferred cells. When we assessed the metabolic activity of various CD8 subpopulations at d15 and 21 p.i. using SCENITH, we noticed that IRF-5 deficiency was mostly affecting the lipid metabolism in Tex^{int} ($\text{Ly108}^+\text{CD69}^-$) and terminally exhausted ($\text{Ly108}^-\text{CD69}^+$) p14 CD8 T cells, but not $\text{Tex}^{\text{prog1}}$ and $\text{Tex}^{\text{prog2}}$. These findings are now shown in Figure 7 A-G.

11) *Reduced TMRM fluorescence does not show "cells that are metabolically stressed"; do you have used uncoupling agents as a negative control?*

We did not use uncoupling agents as a negative control. We have now removed that statement.

12) *5 μ M MitoSOX leads to DNA staining and does not represent mROS, <1 μ M is required! How should mROS (if correctly measured) contribute to this matter? Of note, mROS are required for T cell activation.*

We are thankful to Reviewer #2 for this suggestion. We have now repeated the experiment using the right concentration of MitoSOX and observed a highly significant difference between WT and KO cells at d21p.i.. We have now replaced the old graph with the new one.

13) *Data on lipid long chain fatty acid uptake is incongruent between Fig 6/7, further highlighting the different experimental models.*

We agree that the in vitro model does not entirely recapitulate the in vivo model. For this reason, we have now added new data (s. point 11 above).

14) *The authors do not specify "metabolic rewiring" during chronic infection, which makes it difficult to identify "changes" to metabolic rewiring.*

We also agree with this point. Our interpretation was speculative. We have now changed the title to reflect a more appropriate description of our findings and characterized IRF-5 as a metabolic check point.

15) *How is lipid peroxidation involved in this matter?*

We do not know yet the exact mechanism that leads to increased mitoROS and lipid peroxidation. It was recently demonstrated that mitochondrial dysfunction is responsible for the transition from progenitor-like cells to terminal exhaustion (Wu et al., Nat Comm 2023). This dysfunction causes redox stress, which leads to HIF-1 α stabilization causing glycolytic reprogramming of the cellular metabolism and pushing cells towards terminal exhaustion.

We think that the IRF-5 not only slows down the transition towards exhaustion by sustaining the lipid metabolism, but it also supports mitochondria to adapt to the continue antigen stimulation and prevent damage. It is possible that IRF-5 may participate to the oxidative stress response. We are planning to explore this possibility in future studies.

16) *Please specify the concept of "mitochondrial envelope".*

We have now removed this data from the manuscript.

17) *Please clearly state in the abstract and in the introduction that only murine cells are being investigated.*

We have modified the abstract; in the last paragraph of the introduction, we had already mentioned that LCMV infection was done in mice.

18) *Several pages of the results describe the names and context of transcripts that are irrelevant for the study, please check and remove to keep the overall view.*

We have modified the text accordingly.

19) *Since the role of IRF5 in cell cycle arrest has already been described, it remains elusive why the authors examine this subject again and provide a complete figure about it.*

We have provided a full figure because transcription factors may have different functions in different cell types. We believe that the observation that cells enter the cell cycle more rapidly in the absence of IRF-5 but fail to accumulate is important to interpret the full story.

20) *Given the tremendous amount of transcription factors, it should be clarified what the value of IRF5 is in the overall concept of CD8 T cell exhaustion.*

We hoped that our discussion helped to clarify this point. We think that IRF-5 acts as a metabolic check point, in that it slows down the progress towards terminal exhaustion by supporting the oxidative phosphorylation and mitochondrial function and limiting cell division. We believe that these IRF-5 functions are mostly occurring in effector and terminally exhausted cells rather than in progenitor-like cells, where it may play a very different role.

IRF-5 also competes for DNA binding with IRF-4 (and indirectly HIF-1 α), which is known to promote CD8 T cell exhaustion. However, IRF-5 functions are certainly not limited to the competition with IRF-4. For example, Blimp-1 is a known downstream target of IRF-5 in other cell types and is a regulator of energy metabolism in CD8 T cells. We also see an increase in TOX expression in cells lacking IRF-5. At the moment, we do not know the relationship between TOX and IRF-5. Moreover, we are currently investigating another possible downstream target that is involved in modulating the lipid metabolism. The current study is just an initial observation. Further studies are definitely warranted to better understand the contribution of IRF-5 in the context of other transcription factors.

21) *The introduction is rather a review article itself and should be significantly shortened.*

We have shortened the introduction as suggested.

Reviewer#3

Mai and colleagues analyze the role played by the nuclear factor IRF5 in the phenomenon of T cell exhaustion following infection with clone 13 LCMV. They present data supporting a role played by IRF5 in the limitation of T cell functional exhaustion. Though TcR-transgenic virus-specific IRF5-deficient CD8 T cells exhibited a better proliferative capacity in the acute phase of infection, they were prone to apoptosis, exhibited deeper exhaustion in the chronic phase and were unable to control virus replication. They also provide evidence, mostly based on gene expression analysis and by in vitro metabolic inhibition, that the absence of IRF5 expression limits glycolysis, promote oxidative stress accompanied by lipid peroxidation.

One of the weaknesses of this work is that it is very descriptive and does not present an in-depth description of the mechanisms underlying IRF5 expression.

I have a few remarks that need authors' attention:

1) *Legend of Figure 1 must be modified. In the legend it is said that "Graph shows ... the frequency (right) of p14 CD8 T cells present in the spleen of infected mice." To be correct, it*

should be : "Graph shows ... the frequency (right) of p14 CD8 T cells expressing IRF5 present in the spleen of infected mice."

We thank Reviewer #3 for the correction. The figure legend has been amended.

2) page 6, first paragraph of the results section, it is said that "The kinetics of IRF-5 expression in CD127+ CD8+ T cells was slightly different. Indeed, higher frequencies of IRF-5+ CD127+ cells were observed between d8 and 34p.i. (Figure 1C)." This is not the case since at day 15 the percentage of IRF5-expressing cells among CD127+ cells is half that at day 8. Please modify. Thank you. We modified the text.

3) Same page, same paragraph, it is also said "IRF-5 expression was found to co-localize with the nucleus in CD8 T cells during the whole course of infection, with 60-70% of the cells expressing IRF-5 in the nucleus between d8 and 60 p.i. (Figure 1D). " When I look at the graph, I only see 50 to 70%. Please modify.

4) Page 7: "To determine the function of IRF-5 in CD8 T cells, we generated IRF-5-deficient p14 transgenic mice (*Irf5*^{fl/flx} CMV-Cre⁺ p14), which have CD8 T cells that are specific for the LCMV gp33 antigen". To be fair, it should be specified, here or in the Mat and Methods, that, because of Cre being expressed under CMV promoter, ablation of IRF5 expression in these mice is ubiquitous and not restricted to T cells. The origin of p14 mice should also be mentioned.

We obtained our mice from Dr. Alain Lamarre, a co-author.

We have modified the text to clarify that the *Irf5*^{fl/flx} CMV-Cre⁺ are total KO mice.

5) The authors observed a difference of survival after infection between mice transferred with p14 T cells from male or female donors (Fig S1). Since, in C57BL/6 mice, females develop immunity against male antigens, it should be specified whether p14 T cells were transferred in sex-matched recipients.

We have now specified this in the Material and Methods section.

6) I am a bit concerned about the experiments analyzing the metabolism of IRF-/- p14 CD8 T cells. These experiments were carried out after stimulating in vitro naive T cells with specific antigenic peptide. Results show that IRF-/- CD8 T cells displayed deeply impaired glycolytic and oxidative metabolisms 24 hr after stimulation (see Figure 6). This is surprising to me since increased metabolism is important for T cell proliferation and, according to Figure 2C, the same cells appear to have no problem expanding in the acute phase of LCMV infection (3000 p14 T cells were injected per mouse and more than 7 millions of them were recovered 8 days after infection). Can the authors comment on this. Have the authors analyzed naive IRF-/- p14 CD8 T cells for in vitro proliferation after peptide stimulation?

We agree that our in vitro results are in disagreement with the unaltered clonal expansion of IRF-5-deficient p14 cells observed in vivo. Cells in vivo will receive other signals besides TCR/MHC-I and these may lead to compensatory pathways; moreover, IRF-4, who typically competes for DNA binding with IRF-5, supports glycolysis. Because of this discrepancy, we assessed the cell metabolic activity of the various CD8 T cell subpopulations at d15 and 21 p.i. ex-vivo using the

SCENITH kit (Arguello et al, Cell Metabolism 2020). This method indirectly monitors the metabolic activity of the cells by assessing their translation level. The results have been added in Figure 7 (A-G). Our new data shows that IRF-5 deficiency mostly affects the lipid metabolism in Tex^{int} ($\text{Ly108}^+\text{CD69}^-$) and terminally exhausted ($\text{Ly108}^-\text{CD69}^+$) p14 CD8 T cells, but not in $\text{Tex}^{\text{prog1}}$ and $\text{Tex}^{\text{prog2}}$. We did not see any difference in the glycolytic activity in the absence of IRF-5,

7) *Fig 5A and 5B display expression of Cell Death genes and Exhaustion genes, respectively. It is the opposite in the legend of the figure. Please correct.*

We have now removed those GSEA from Figure 5.

8) *Since analysis of mitochondrial ROS production did not show any significant differences between IRF5^{-/-} and WT T cells (Fig 7E), the data should not be presented as supporting the presence of mitochondrial stress in IRF5^{-/-} T cells (page 16).*

We apologize for the confusion. Reviewer #2 noticed that we did not use the right concentration of MitoSOX. We have now repeated the experiment as advised and KO cells have significantly higher MtSOX MFI than WT cells.

9) *Please check the references. In the discussion section page 18, line 4, reference #58 is given for the role of TOX in T cell development while the study presented there talks about the assembly of mitochondrial complex 1.*

Thank you. We changed the reference.

Dear Prof. Stager,

Thank you for submitting your revised manuscript EMBOJ-2024-117192R-Q for consideration by The EMBO Journal, and for your patience during peer review. Your manuscript has now been seen by the three referees that had also previously assessed the initial version of your manuscript, and we have received the full set of their comments, which you can find below.

As you will see, all three referees recognize that the majority of the initially raised concerns have been satisfactorily addressed in this thoroughly revised and substantially strengthened version of your manuscript. There is a number of remaining concerns and suggestions for further improvement, however, which we would like you to address in a final revision. In particular, referees #1 and #2 both raise the concern that the available data do not conclusively prove that the observed cell death in the context of your study is mediated by lipid peroxidation-induced ferroptosis, and they suggest that your claim be either validated further (e.g. by using ferroptosis inhibitors) or removed. Referee #2 lists a number of additional suggestions for strengthening the study and the manuscript further, which we agree would increase the impact of your work on the field. Some of these points are optional and will not be required for the publication of your study in The EMBO Journal, but those ones that relate to the solidity of the conclusions and the clarity of their presentation in the manuscript must be addressed. You are also kindly requested to address the remaining remark of referee #3.

Given the referees' positive comments and recommendations, I would like to invite you to submit a final version of the manuscript along with a detailed point-by-point response addressing all referees' comments. Please explain in your response which points (and how) are addressed in this version, and which (and why) you cannot address. I should add that acceptance of your manuscript will depend on the completeness of your responses in this revised version. Please let me know if you have any questions or comments that you would like to discuss with me.

We generally allow three months as standard revision time (May 4, 2025). As a matter of policy, competing manuscripts published during this period will not negatively impact our assessment of the conceptual advance presented by your study. However, we request that you contact us as soon as possible upon publication of any related work, to discuss how to proceed. Should you foresee a problem in meeting this three-month deadline, please let us know in advance and we may be able to grant an extension.

Thank you for the opportunity to consider your work for publication in The EMBO Journal. I look forward to your revision.

Best regards,

Ioannis

Instructions for preparing your revised manuscript

1. When you are ready to submit the revision, please upload:

- A Word file of the manuscript text (including legends of main Figures, EV Figures and Tables). Please make sure that changes are highlighted (or "tracked") to be clearly visible.

- Individual production-quality figure files (one file per figure). When assembling your figures, please refer to our figure preparation guidelines in order to ensure proper formatting and readability in print as well as on screen:

If the data shown in a figure are obtained from n {less than or equal to} 2, please use scatter plots showing the individual data points.

- i. the name of the statistical test used to generate error bars and P values
- ii. the number (n) of independent experiments (please specify technical or biological replicates) underlying each data point (discussion of statistical methodology can be reported in the Materials and Methods section, but figure legends should contain a basic description of n , P, and the test applied)

iii. the nature of the bars and error bars (s.d., s.e.m.).

- A point-by-point response to the referees' comments, with a detailed description of the changes made (as a word file). All referees' concerns must be fully addressed and their suggestions taken on board. When preparing your letter of response to the referees' comments, please bear in mind that this will form part of the Review Process File and will therefore be available online to the community. Please note that you have the possibility to opt out of the transparent process at any stage prior to publication by letting the editorial office know (contact@embojournal.org); if you do opt out, the Review Process File link will point to the following statement: "No Review Process File is available with this article, as the authors have chosen not to make the review process public in this case.". For more details on our Transparent Editorial Process, please visit our website: <https://www.embopress.org/page/journal/14602075/authorguide#transparentprocess>

- Expanded View (EV) files (replacing Supplementary Information) that are collapsible/expandable online. A maximum of 5 EV Figures can be typeset. EV Figures should be cited as "Figure EV1, Figure EV2" etc. in the text, and their respective legends should be included in the manuscript file after the legends of regular figures. See detailed instructions regarding Expanded View files here:

- For the figures that you do NOT wish to display as Expanded View figures, they should be bundled together with their legends in a single PDF file called "Appendix", which should start with a short Table of Contents (including page numbers). Appendix figures should be referred to in the main text as: "Appendix Figure S1, Appendix Figure S2" etc. Please see detailed instructions here: <https://www.embopress.org/page/journal/14602075/authorguide#expandedview>

- A complete author checklist, which you can download from our author guidelines (<https://www.embopress.org/page/journal/14602075/authorguide>). Please note that the checklist will also be part of the Review Process File.

2. Please note that no statistics should be calculated and shown in Figures if $n=2$. Please also note that each p value should be reported as an exact value.

3. Before submitting your revision, primary datasets (and computer code, where appropriate) produced in this study need to be deposited in appropriate public databases (see <https://www.embopress.org/page/journal/14602075/authorguide#dataavailability>).

In particular, you are kindly requested to deposit all RNA-seq datasets produced in your study in an appropriate database. The accession numbers, database, and the specific URLs (links) should be listed in a formal "Data availability" section (placed after Methods), following the example below:

"The RNA-seq datasets produced in this study are available in the following database:
Gene Expression Omnibus GSE46843 (<https://www.ncbi.nlm.nih.gov/geo/query/acc.cgi?acc=GSE46843>)"

*** All links should resolve to a page where the data can be accessed. ***

*** Please remember to provide in the Data availability section of your revised manuscript reviewer passwords if the datasets are not yet public. ***

*** The Data Availability Section is restricted to new primary data that are part of this study. In case you have no data that require deposition in a public database, please state so instead of referring to the database: "Our study includes no data deposited in public repositories." under the heading "Data availability". ***

4. Please check that the title and the abstract of the manuscript are brief, yet explicit, even to non-specialists. The length of the title should not exceed 100 characters, and the abstract should be a single paragraph not exceeding 175 words.

5. Please also note our reference format: <https://www.embopress.org/page/journal/14602075/authorguide#referencesformat>.

7. Please remember: digital image enhancement is acceptable practice, as long as it accurately represents the original data and conforms to community standards. If a figure has been subjected to significant electronic manipulation, this must be noted in the figure legend or in the "Materials and Methods" section. The editors reserve the right to request original versions of figures and the original images that were used to assemble the figure.

8. Our journal encourages inclusion of data citations in the reference list to directly cite datasets that were obtained from public

databases. Data citations in the article text are distinct from normal bibliographical citations and should directly link to the database records from which the data can be accessed. In the main text, data citations are formatted as follows: "Data ref: Smith et al, 2001" or "Data ref: NCBI Sequence Read Archive PRJNA342805, 2017". In the Reference list, data citations must be labeled with "[DATASET]". A data reference must provide the database name, accession number/identifiers, and a resolvable link to the landing page from which the data can be accessed at the end of the reference. Further instructions are available at: <https://www.embopress.org/page/journal/14602075/authorguide#referencesformat>.

9. We request authors to consider both actual and perceived competing interests. Please review our policy (<https://www.embopress.org/page/journal/14602075/authorguide#conflictsinterest>) and update your competing interests statement if necessary. Please name this section 'Disclosure and competing interests statement' and place it after the Acknowledgements section.

10. Please note that all corresponding authors are required to provide an ORCID ID upon submission of a revised manuscript (<https://orcid.org/>). Please find instructions on how to link your ORCID ID to your account in our manuscript tracking system in our Author guidelines (<https://www.embopress.org/page/journal/14602075/authorguide#authorshipguidelines>).

11. We use CRediT to specify the contributions of each author in the journal submission system. CRediT replaces the author contribution section, which should be removed from the manuscript. Please use the free text box to provide more detailed descriptions. See also guide to authors: <https://www.embopress.org/page/journal/14602075/authorguide#authorshipguidelines>.

13. We would also welcome the submission of cover suggestions or motifs to be used by our Graphics Illustrator in designing a cover.

14. Please use the link below to submit your revision:
<https://emboj.msubmit.net/cgi-bin/main.plex>

Referee #1:

The authors have address most of the concerns raised by this reviewer. However, the abstract needs to be adjusted. Currently the abstract finishes with the sentence "These findings identify IRF-5 as a pivotal metabolic checkpoint in CD8 T cells during the chronic stages of infection and highlight its role in protecting cells from cell death by lipid peroxidation-induced ferroptosis". The authors do not provide any evidence that the CD8 T cell die by ferroptosis.

Referee #2:

The authors present a major update of their initial submission, now focussing on the role of the transcription factor IRF-5 as an essential metabolic checkpoint in CD8 T cells during chronic infection. This manuscript now includes many new data, figures and methods. The authors have considered most of the suggestions of all three reviewers. This revised version has substantially improved.

Especially the new data on ex vivo metabolism using Scenith and TEM analyses significantly improve the understanding of the role of IRF5. I have a few suggestions to the authors.

- The changes in mitochondrial morphology are very important. I would strongly recommend discussing the potential link between these alterations and the metabolic disturbances
- Also related to this issue: you might consider quantification of mean cristae width, as this has been proposed as an indicator of OXPHOS efficiency and might provide an additional explanation to your findings (PMID: 24055366)
- Fig 7A suggests that ex vivo metabolic capacity (as indicate by translational capacity) does not change in IRF5^{-/-}. Do you have ECAR data related to Fig 6 that might backup this (or show discrepancies)?
- The authors report increased mitochondrial mass with reduced mitochondrial membrane integrity. This might indicate the induction of -yet insufficient- mitochondrial biogenesis in IRF5^{-/-}. This could be analyzed via TFAM quantification.
- Is the increase of mROS accompanied by a reduction in antioxidative capacity? This could me quantified by using ThioTracker dye.

- Since now the correct concentration of MitoSOX is used, I strongly suggest correcting this part of the method section as well
- The authors detect an increase in lipid peroxidation; this might occur in response to mROS, or it might primarily affect non-mitochondrial lipids; to gain a first insight into this, it could be useful to also quantify cellular ROS using CellROX
- Have the authors tried to dampen mROS by supplementing antioxidants and thus to restore the mitochondrial function?
- The authors quantify increased CD36 and state that "cells most likely die by ferroptosis"; this is not backed up by the data and I strongly suggest to clearly indicate the speculative character of this statement
- To clearly link ferroptosis to your data, you could use inhibitors of ferroptosis (e.g. ferrostatin)
- Considering the previous and all of the new data, it is tempting to speculate that IRF5 is mainly involved in metabolic or mitochondrial gene expression, with all the effects on cell cycle and exhaustion being a consequence of this. This is also supported by your RNAseq data. Do you have data on specific mitochondrial transcripts using qRT-PCR or have you quantified OXPHOS protein complexes? Ban et al have shown that IRF5 is involved in OXPHOS gene expression (PMID: 34282144)
- Orliaguet et al report that IRF5 deficiency induces GHITM which impacts mitochondrial architecture; as you have also discovered severe changes to mitochondrial structure - do you have quantified GHITM in your setting?
- Could you discuss on why IRF5 deficiency in the abovementioned study led to an increase of OXPHOS compared to the substantial defects displayed in your study?
- In consequence of these points, I would advise to update the discussion and focus on the role of IRF5 as a "metabolic checkpoint" including all the new data, as already indicated by the new title of your manuscript. This might be the far greater scientific advancement as it highlights a substantial role of mitochondrial function for T-cell exhaustion.

Referee #3:

The authors have responded satisfactorily to most of my remarks.

Mino remark: In Figure 7 and text (page 15), unlike the Seahorse which, by measuring the extracellular acidification rate, allows to evaluate the glycolytic activity of the cells, the Scenith technique does not allow it. Indeed, blocking the entry of glucose by 2-DG does not say whether protein synthesis will be supported by glycolysis (production of lactate- + H+) or glucose oxidation (OXPHOS). "% of glycolysis-dependent translational level" should be replaced by "% of glucose-dependent translational level".

We are very grateful for the reviewers' insightful comments, which we have addressed as follows.

Editor's suggestions: *"As you will see, all three referees recognize that the majority of the initially raised concerns have been satisfactorily addressed in this thoroughly revised and substantially strengthened version of your manuscript. There is a number of remaining concerns and suggestions for further improvement, however, which we would like you to address in a final revision. In particular, referees #1 and #2 both raise the concern that the available data do not conclusively prove that the observed cell death in the context of your study is mediated by lipid peroxidation-induced ferroptosis, and they suggest that your claim be either validated further (e.g. by using ferroptosis inhibitors) or removed. Referee #2 lists a number of additional suggestions for strengthening the study and the manuscript further, which we agree would increase the impact of your work on the field. Some of these points are optional and will not be required for the publication of your study in The EMBO Journal, but those ones that relate to the solidity of the conclusions and the clarity of their presentation in the manuscript must be addressed. You are also kindly requested to address the remaining remark of referee #3."*

Reviewer#1

"The authors have address most of the concerns raised by this reviewer. However, the abstract needs to be adjusted. Currently the abstract finishes with the sentence "These findings identify IRF-5 as a pivotal metabolic checkpoint in CD8 T cells during the chronic stages of infection and highlight its role in protecting cells from cell death by lipid peroxidation-induced ferroptosis". The authors do not provide any evidence that the CD8 T cell die by ferroptosis."
We agree with reviewer#1. This conclusion is speculative. We have now modified the text to read *"These findings identify IRF-5 as a pivotal metabolic checkpoint in CD8 T cells during the chronic stages of infection and highlight its role in regulating mitochondria remodelling and protecting cells from exhaustion"*.

Reviewer#2

"The authors present a major update of their initial submission, now focussing on the role of the transcription factor IRF-5 as an essential metabolic checkpoint in CD8 T cells during chronic infection. This manuscript now includes many new data, figures and methods. The authors have considered most of the suggestions of all three reviewers. This revised version has substantially improved. Especially the new data on ex vivo metabolism using Scenith and TEM analyses significantly improve the understanding of the role of IRF5. I have a few suggestions to the authors."

We thank Reviewer#2 for the appreciation of our work.

"The changes in mitochondrial morphology are very important. I would strongly recommend discussing the potential link between these alterations and the metabolic disturbances"
"Also related to this issue: you might consider quantification of mean cristae width, as this has been proposed as an indicator of OXPHOS efficiency and might provide an additional explanation to your findings (PMID: 24055366)"

We have now assessed the mean cristae width and found that IRF-5-deficient cells have a significantly larger mean cristae width compared with IRF-5 sufficient cells, which is in agreement with the observed defects in mitochondrial functions. Thank you for this excellent suggestion. These results are now presented in Figure 6N and we have updated the discussion.

“Fig 7A suggests that ex vivo metabolic capacity (as indicated by translational capacity) does not change in IRF5^{-/-}. Do you have ECAR data related to Fig 6 that might backup this (or show discrepancies)?”

We have now added the ECAR data for the in vitro stimulation (Supplemental Figure 7A-B for female mice and Supplemental Figure 8C-D for male mice).

“The authors report increased mitochondrial mass with reduced mitochondrial membrane integrity. This might indicate the induction of -yet insufficient- mitochondrial biogenesis in IRF5^{-/-}. This could be analyzed via TFAM quantification.”

This is an interesting suggestion; however, we did not analyze TFAM expression because we did not see any difference in our transcriptomic analysis

Mitochondrial transcription factor A (TFAM)

Day_21_male_WT1	1029.828	Day_21_male_KO1	902.5219
Day_21_male_WT4	1077.495	Day_21_male_KO4	903.6665
Day_21_female_WT2	947.0615	Day_21_female_KO2	906.1518
Day_21_female_WT3	1100.206	Day_21_female_KO3	1150.068

“Is the increase of mROS accompanied by a reduction in antioxidative capacity? This could be quantified by using ThioTracker dye”. “Have the authors tried to dampen mROS by supplementing antioxidants and thus to restore the mitochondrial function?”

We thank Reviewer#2 for the great suggestions. We are actually planning to investigate the oxidative stress response and evaluate in detail the antioxidative capacity of IRF-5-deficient cells. This will be an important topic for future studies.

“Since now the correct concentration of MitoSOX is used, I strongly suggest correcting this part of the method section as well”

We change this, thank you.

“The authors detect an increase in lipid peroxidation; this might occur in response to mROS, or it might primarily affect non-mitochondrial lipids; to gain a first insight into this, it could be useful to also quantify cellular ROS using CellROX”

Another great suggestion, thank you. We are also planning to further investigate the pathways leading to cell death in the absence of IRF-5 and a more thorough analysis of ROS production including cellular ROS will definitely be part of this. We have now removed the link between cell death and lipid peroxidation, so measuring cellular ROS, although very important in general, becomes more secondary to our current study.

“The authors quantify increased CD36 and state that “cells most likely die by ferroptosis”; this is

not backed up by the data and I strongly suggest to clearly indicate the speculative character of this statement” “To clearly link ferroptosis to your data, you could use inhibitors of ferroptosis (e.g. ferrostatin)”

We agree with Reviewer#2 that our statement was speculative. As suggested by the Editor, we have now decided to remove the CD36 expression analysis and the conclusion that cells die by ferroptosis.

“Considering the previous and all of the new data, it is tempting to speculate that IRF5 is mainly involved in metabolic or mitochondrial gene expression, with all the effects on cell cycle and exhaustion being a consequence of this. This is also supported by your RNAseq data. Do you have data on specific mitochondrial transcripts using qRT-PCR or have you quantified OXPHOS protein complexes? Ban et al have shown that IRF5 is involved in OXPHOS gene expression (PMID: 34282144)”

We had some digital droplet PCR data on the mRNA expression of genes involved in mitochondrial and metabolic gene expression. These have now been added in Figures 5G and 5K.

“Orliaguet et al report that IRF5 deficiency induces GHITM which impacts mitochondrial architecture; as you have also discovered severe changes to mitochondrial structure - do you have quantified GHITM in your setting?”

Yes, we did, and we also see an increase in *Ghitm* mRNA expression. These results are now shown in Figure 5H.

“Could you discuss on why IRF5 deficiency in the abovementioned study led to an increase of OXPHOS compared to the substantial defects displayed in your study?”

We have now added a paragraph discussing the discrepancy between our study and the work by Orliaguet et al.

“In consequence of these points, I would advise to update the discussion and focus on the role of IRF5 as a "metabolic checkpoint" including all the new data, as already indicated by the new title of your manuscript. This might be the far greater scientific advancement as it highlights a substantial role of mitochondrial function for T-cell exhaustion”

We have now updated the discussion and highlighted the importance of the role of IRF-5 in remodelling the mitochondria and how this influences the mitochondria respiratory efficiency. Thank you for this great suggestion.

Reviewer #3

“The authors have responded satisfactorily to most of my remarks. Mino remark: In Figure 7 and text (page 15), unlike the Seahorse which, by measuring the extracellular acidification rate, allows to evaluate the glycolytic activity of the cells, the Scenith technique does not allow it. Indeed, blocking the entry of glucose by 2-DG does not say whether protein synthesis will be supported by glycolysis (production of lactate- + H+) or glucose

oxidation (OXPHOS). "% of glycolysis-dependent translational level" should be replaced by "% of glucose-dependent translational level".

Thank you for the correction. We have now changed the graph labeling as suggested.

Dear Simona,

Thank you again for submitting your revised manuscript (EMBOJ-2024-117192R1) to The EMBO Journal for our consideration, and for your patience during peer review. We have shared the revised version of your manuscript with referee #2, and we have now received their comments, which you can find below. I am very pleased to say that the referee recognizes that the manuscript has been significantly strengthened and their previous concerns have been adequately and sufficiently addressed. In light of this input, I am glad to inform you that your manuscript has now been in principle accepted for publication in The EMBO Journal - congratulations on an excellent study.

Before we can proceed with formal acceptance of the manuscript and its publication, however, there are several changes and corrections from the editorial side we need you to make in a final version of the manuscript:

- Please note that "equal contribution" is allowed only for the first (co-first) or corresponding (co-corresponding authors). In your author list, you indicate two "co-second" co-authors, which is not possible. You are kindly requested to revise the author list according to the contributions of each co-author to the study and the manuscript; those must be specified, for all co-authors, in detail using the CRediT author contribution taxonomy system during submission of the final version of your manuscript.
- Please note that all funding information provided in the Acknowledgements section of your manuscript must also be entered in our manuscript tracking system during submission of the final version of the manuscript. Currently, the listed "scholarships from the Fondation Armand-Frappier and the Fonds de Recherche du Québec - Santé (FRQS)" are missing from our online system.
- Please provide a list of up to 5 relevant keywords after the Abstract of your revised manuscript - those will be important for the search engine discoverability of your paper.
- We noticed that the reference format is not compatible with our journal's style (please note that the names of the first 10 co-authors should be listed for each reference, followed by "et al." in cases of papers with more than 10 co-authors) - please find more information on the required reference format in our guide below and update your references accordingly:
<https://www.embopress.org/page/journal/14602075/authorguide#referencesformat>.
- Thank you for providing reviewer access to your deposited datasets. The confidential token can now be removed from your "Data availability" statement. Please make sure that all datasets will be publicly available at the time of publication, and provide the specific links (URLs) in the "Data availability" statement.
- Please change the heading of your "Competing interests" statement to "Disclosure and competing interests statement".
- The author contributions statement should be removed from the manuscript file. Instead, we use CRediT to specify the contributions of each author in the journal submission system. Please feel free to use the free text box to provide more detailed descriptions during submission. See also our guide to authors for more information:
<https://www.embopress.org/page/journal/14602075/authorguide#authorshippinguidelines>.
- All Figure callouts should be listed sequentially.
- Figures should be uploaded as individual Figure files; for more information on Figure formatting please see our guide:
<https://www.embopress.org/page/journal/14602075/authorguide#figureformat>.
- The Appendix file needs to be in PDF format; its title page should begin with "Appendix for" followed by the manuscript title, and a Table of Contents with the page numbers of the listed items; the nomenclature of the listed items should be "Appendix Figure S#" and "Appendix Table S#" throughout the main manuscript and the Appendix PDF file; the Appendix Figure legends should be removed from the main manuscript file and only be included in the Appendix file; Supplemental Table 1 should be included in the Appendix PDF file with the correct nomenclature; Supplemental Table 2 (primer information) should be incorporated in the Reagents and Tools Table.
- Please make sure to upload all requested Source Data for your Figures, or clearly explain in your Source Data checklist where these data can be accessed from (if, for example, some of these data are included in the BioStudies accession listed in your Data availability statement). Please make sure that the access information to all Source Data that are not uploaded to our manuscript tracking system must be included in the Data availability statement.
- Please note that EMBO press papers are accompanied online by:
 - A) a short (2 sentences) summary of the findings and their significance,
 - B) 2-5 short bullet points highlighting the key results, and
 - C) a synopsis image in .jpg or .png format that is exactly 550 pixels wide and 300-600 pixels high (the height is variable). Please note that the text needs to be legible at the final size.

Please upload this information along with your revised manuscript (the text for A and B should be provided in a separate Word file).

- The order of the manuscript sections must be corrected as follows: Title page - Abstract and Keywords - Introduction - Results - Discussion - Methods - Data Availability - Acknowledgements - Disclosure and Competing Interests Statement - References - Figure Legends - main Tables (if there are any) - Expanded View Figure Legends.

Please also note that as part of the EMBO publications' Transparent Editorial Process, The EMBO Journal publishes online a Peer Review File along with each accepted manuscript. This File will be published in conjunction with your paper and will include the referee reports, your point-by-point response and all pertinent correspondence relating to the manuscript. You can opt out of this by letting the editorial office know (contact@embojournal.org). If you do opt out, the Peer Review File link will point to the following statement: "No Peer Review File is available with this article, as the authors have chosen not to make the review process public in this case."

We look forward to seeing a final version of your manuscript as soon as possible. Please let us know if you have any questions and use this link to submit your revision: <https://emboj.msubmit.net/cgi-bin/main.plex>.

Best wishes,

Ioannis

Referee #2:

The authors have again thoroughly revised and improved their manuscript. My concerns and suggestions have been addressed adequately and I congratulate the authors to this final version of their manuscript.

All editorial and formatting issues were resolved by the authors.

Dear Simona,

Congratulations on an excellent study! I am very pleased to inform you that your manuscript has now been accepted for publication in The EMBO Journal. Thank you very much for addressing the referees' concerns and the editorial requests for changes.

If you have any questions, please do not hesitate to contact the Editorial Office. Thank you for your contribution to The EMBO Journal. Working with you has been a pleasure.

Best wishes,

Ioannis
